# You Only Condense Once: Two Rules for Pruning Condensed Datasets

**Yang He, Lingao Xiao, Joey Tianyi Zhou**[✉]
CFAR, Agency for Science, Technology and Research, Singapore
IHPC, Agency for Science, Technology and Research, Singapore
`{He_Yang, Joey_Zhou}@cfar.a-star.edu.sg`

## Abstract

Dataset condensation is a crucial tool for enhancing training efficiency by reducing the size of the training dataset, particularly in on-device scenarios. However, these scenarios have two significant challenges: 1) the varying computational resources available on the devices require a dataset size different from the pre-defined condensed dataset, and 2) the limited computational resources often preclude the possibility of conducting additional condensation processes. We introduce **You Only Condense Once (YOCO)** to overcome these limitations. On top of one condensed dataset, YOCO produces smaller condensed datasets with two embarrassingly simple dataset pruning rules: **Low LBPE Score** and **Balanced Construction**. YOCO offers two key advantages: 1) it can flexibly resize the dataset to fit varying computational constraints, and 2) it eliminates the need for extra condensation processes, which can be computationally prohibitive. Experiments validate our findings on networks including ConvNet, ResNet and DenseNet, and datasets including CIFAR-10, CIFAR-100 and ImageNet. For example, our YOCO surpassed various dataset condensation and dataset pruning methods on CIFAR-10 with ten Images Per Class (IPC), achieving 6.98-8.89% and 6.31-23.92% accuracy gains, respectively. The code is available at: `https://github.com/he-y/you-only-condense-once`.

## 1 Introduction

Deep learning models often require vast amounts of data to achieve optimal performance. This data-hungry nature of deep learning algorithms, coupled with the growing size and complexity of datasets, has led to the need for more efficient dataset handling techniques. Dataset condensation is a promising approach that enables models to learn from a smaller and more representative subset of the entire dataset. Condensed datasets are especially utilized in on-device scenarios, where limited computational resources and storage constraints necessitate the use of a compact training set.

However, these on-device scenarios have two significant constraints. First, the diverse and fluctuating computational resources inherent in these scenarios necessitate a level of flexibility in the size of the dataset. But the requirement of flexibility is not accommodated by the fixed sizes of previous condensed datasets. Second, the limited computational capacity in these devices also makes extra condensation processes impractical, if not impossible. Therefore, the need for adaptability in the size of the condensed dataset becomes increasingly crucial. Furthermore, this adaptability needs to be realized without introducing another computationally intensive condensation process.

We introduce **You Only Condense Once (YOCO)** to enable the **flexible resizing (pruning)** of condensed datasets to fit varying on-device scenarios without extra condensation process (See Fig. 1). The first rule of our proposed method involves a metric to evaluate the importance of training samples in the context of dataset condensation.

---

[✉] Corresponding Author

37th Conference on Neural Information Processing Systems (NeurIPS 2023).

From the gradient of the loss function, we develop the **Logit-Based Prediction Error (LBPE) score** to rank training samples. This metric quantifies the neural network's difficulty in recognizing each sample. Specifically, training samples with low LBPE scores are considered easy as they indicate that the model's prediction is close to the true label. These samples exhibit simpler patterns, easily captured by the model. Given the condensed datasets' small size, prioritizing easier samples with **low LBPE scores** is crucial to avoid overfitting.

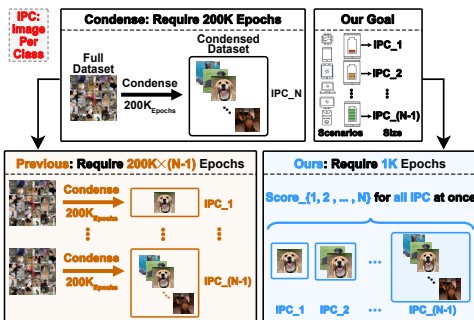

Figure 1: Previous methods (left) require extra condensation processes, but ours (right) do not.

A further concern is that relying solely on a metric-based ranking could result in an imbalanced distribution of classes within the dataset. Imbalanced datasets can lead to biased predictions, as models tend to focus on the majority class, ignoring the underrepresented minority classes. This issue has not been given adequate attention in prior research about dataset pruning [3, 41, 34], but it is particularly important when dealing with condensed datasets. In order to delve deeper into the effects of class imbalance, we explore Rademacher Complexity [31], a widely recognized metric for model complexity that is intimately connected to generalization error and expected loss. Based on the analysis, we propose **Balanced Construction** to ensure that the condensed dataset is both informative and balanced.

The key contributions of our work are: **1)** To the best of our knowledge, it's the first work to provide a solution to adaptively adjust the size of a condensed dataset to fit varying computational constraints. **2)** After analyzing the gradient of the loss function, we propose the LBPE score to evaluate the sample importance and find out easy samples with low LBPE scores are suitable for condensed datasets. **3)** Our analysis of the Rademacher Complexity highlights the challenges of class imbalance in condensed datasets, leading us to construct balanced datasets.

## 2 Related Works

### 2.1 Dataset Condensation/Distillation

Dataset condensation, or distillation, synthesizes a compact image set to maintain the original dataset's information. Wang *et al.* [45] pioneer an approach that leverages gradient-based hyper-parameter optimization to model network parameters as a function of this synthetic data. Building on this, Zhao *et al.* [53] introduce gradient matching between real and synthetic image-trained models. Kim *et al.* [19] further extend this, splitting synthetic data into $n$ factor segments, each decoding into $n^2$ training images. Zhao & Bilen [50] apply consistent differentiable augmentation to real and synthetic data, thus enhancing information distillation. Cazenavette *et al.* [1] propose to emulate the long-range training dynamics of real data by aligning learning trajectories. Liu *et al.* [24] advocate matching only representative real dataset images, selected based on latent space cluster centroid distances. Additional research avenues include integrating a contrastive signal [21], matching distribution or features [52, 44], matching multi-level gradients [16], minimizing accumulated trajectory error [7], aligning loss curvature [39], parameterizing datasets [6, 23, 51, 43], and optimizing dataset condensation [27, 32, 33, 42, 55, 25, 49, 4, 26]. Nevertheless, the fixed size of condensed datasets remains an unaddressed constraint in prior work.

### 2.2 Dataset Pruning

Unlike dataset condensation that alters image pixels, dataset pruning preserves the original data by selecting a representative subset. Entropy [3] keeps hard samples with maximum entropy (uncertainty [22, 38]), using a smaller proxy network. Forgetting [41] defines forgetting events as an accuracy drop at consecutive epochs, and hard samples with the most forgetting events are important. AUM [35] identifies data by computing the Area Under the Margin, the difference between the true label logits and the largest other logits. Memorization [11] prioritizes a sample if it significantly improves the probability of correctly predicting the true label. GraNd [34] and EL2N [34] classify samples as hard based on the presence of large gradient norm and large norm of error vectors, respectively. CCS [54] extends previous methods by pruning hard samples and using stratified sampling to

achieve good coverage of data distributions at a large pruning ratio. Moderate [47] selects moderate samples (neither hard nor easy) that are close to the score median in the feature space. Optimization-based [48] chooses samples yielding a strictly constrained generalization gap. In addition, other dataset pruning (or coreset selection) methods [8, 28, 36, 17, 30, 18, 2, 29, 9, 10, 46, 15] are widely used for Active Learning [38, 37]. However, the previous methods consider full datasets and often neglect condensed datasets.

## 3 Method

### 3.1 Preliminaries

We denote a dataset by $S = (x_i, y_i)_{i=1}^{N}$, where $x_i$ represents the $i^{th}$ input and $y_i$ represents the corresponding true label. Let $\mathcal{L}(p(\mathbf{w}, x), y)$ be a loss function that measures the discrepancy between the predicted output $p(\mathbf{w}, x)$ and the true label $y$. The loss function is parameterized by a weight vector $\mathbf{w}$, which we optimize during the training process.

We consider a time-indexed sequence of weight vectors, $\mathbf{w}_t$, where $t = 1, \ldots, T$. This sequence represents the evolution of the weights during the training process. The gradient of the loss function with respect to the weights at time $t$ is given by $g_t(x, y) = \nabla_{\mathbf{w}_t} \mathcal{L}(p(\mathbf{w}_t, x), y)$.

### 3.2 Identifying Important Training Samples

Our goal is to propose a measure that quantifies the importance of a training sample. We begin by analyzing the gradient of the loss function with respect to the weights $\mathbf{w}_t$:

$$\nabla_{\mathbf{w}_t} \mathcal{L}(p(\mathbf{w}_t, x), y) = \frac{\partial \mathcal{L}(p(\mathbf{w}_t, x), y)}{\partial p(\mathbf{w}_t, x)} \cdot \frac{\partial p(\mathbf{w}_t, x)}{\partial \mathbf{w}_t}. \tag{1}$$

We aim to determine the impact of training samples on the gradient of the loss function, as the gradient plays a critical role in the training process of gradient-based optimization methods. Note that our ranking method is inspired by EL2N [34], but we interpret it in a different way by explicitly considering the dataset size $|S|$.

**Definition 1 (Logit-Based Prediction Error - LBPE)**: The Logit-Based Prediction Error (LBPE) of a training sample $(x, y)$ at time $t$ is given by:

$$\text{LBPE}_t(x, y) = \mathbb{E} \left| p\left(\mathbf{w}_t, x\right) - y \right|_2, \tag{2}$$

where $\mathbf{w}_t$ is the weights at time $t$, and $p(\mathbf{w}_t, x)$ represents the prediction logits.

**Lemma 1 (Gradient and Importance of Training Samples)**: The gradient of the loss function $\nabla_{\mathbf{w}_t} \mathcal{L}(p(\mathbf{w}_t, x), y)$ for a dataset $S$ is influenced by the samples with prediction errors.

**Proof of Lemma 1**: Consider two datasets $S$ and $S_{\neg j}$, where $S_{\neg j}$ is obtained by removing the sample $(x_j, y_j)$ from $S$. Let the gradients of the loss function for these two datasets be $\nabla_{\mathbf{w}_t}^{S} \mathcal{L}$ and $\nabla_{\mathbf{w}_t}^{S_{\neg j}} \mathcal{L}$, respectively. The difference between the gradients is given by (see **Appendix** A.1 for proof):

$$\Delta \nabla_{\mathbf{w}_t} \mathcal{L} = \frac{-1}{|S|(|S|-1)} \sum_{(x,y) \in S_{\neg j}} \frac{\partial \mathcal{L}(p(\mathbf{w}_t, x), y)}{\partial p(\mathbf{w}_t, x)} \cdot \frac{\partial p(\mathbf{w}_t, x)}{\partial \mathbf{w}_t} + \frac{1}{|S|} \frac{\partial \mathcal{L}\left(p\left(\mathbf{w}_t, x_j\right), y_j\right)}{\partial p\left(\mathbf{w}_t, x_j\right)} \cdot \frac{\partial p\left(\mathbf{w}_t, x_j\right)}{\partial \mathbf{w}_t} \tag{3}$$

Let us denote the error term as: $e_j = p(\mathbf{w}_t, x_j) - y_j$, the LBPE score for sample $(x_j, y_j)$ is given by $\text{LBPE}_t(x_j, y_j) = \mathbb{E} |e_j|_2$, and the difference in gradients related to the sample $(x_j, y_j)$ can be rewritten as:

$$\frac{1}{|S|} \frac{\partial \mathcal{L}(e_j)}{\partial e_j} \cdot \frac{\partial p(\mathbf{w}_t, x_j)}{\partial \mathbf{w}_t}. \tag{4}$$

If the sample $(x_j, y_j)$ has a lower LBPE score, it implies that the error term $e_j$ is smaller. Let's consider the mean squared error (MSE) loss function, which is convex. The MSE loss function is

---

**Algorithm 1** Compute LBPE score for samples over epochs

---

**Require:** Training dataset $S$ and its size $|S|$, weights $\mathbf{w}_t$, true labels $y$, model's predicted probabilities $p(\mathbf{w}_t, x)$, number of epochs $E$, Epochs with Top-K accuracy
 1: Initialize matrix: LBPE= torch.zeros$((E, |S|))$           ▷ LBPE scores over samples and epochs
 2: Initialize accuracy: ACC= torch.zeros$(E)$             ▷ Track the accuracy over epochs
 3: **for** each epoch $t$ in range $E$ **do**                ▷ Loop through each epoch
 4:     **for** each sample index $i$ in $S$ **do**         ▷ Loop through each sample in the dataset
 5:         Compute error term for sample $i$ at epoch $t$: $e_{i,t} = p(\mathbf{w}_t, x_i) - y_i$
 6:         Compute LBPE score for sample $i$ at epoch $t$ with MSE loss: $\text{LBPE}_{i,t} = \mathbb{E}_{MSE} |e_{i,t}|_2$
 7:     **end for**
 8:     Compute accuracy at epoch $t$: $\text{ACC}_t$
 9: **end for**
10: Top_K $\leftarrow$ argsort(ACC)[-k:]             ▷ Find the epochs with the Top-K accuracy
11: AVG_LBPE $\leftarrow$ mean(LBPE[Top_k, :])           ▷ Average LBPE score over Top-K epochs
12: **return** AVG_LBPE

---

defined as $\mathcal{L}(e_j) = \frac{1}{2}(e_j)^2$. Consequently, the derivative of the loss function $\frac{\partial \mathcal{L}(e_j)}{\partial e_j} = e_j$ would be smaller for samples with smaller LBPE scores, leading to a smaller change in the gradient $\Delta\nabla_{\mathbf{w}_t}\mathcal{L}$.

**Rule 1: For a small dataset, a sample with a lower LBPE score will be more important.** Let $S$ be a dataset of size $|S|$, partitioned into subsets $S_{easy}$ (lower LBPE scores) and $S_{hard}$ (higher LBPE scores).

*Case 1: Small Dataset* - When the dataset size $|S|$ is small, the model's capacity to learn complex representations is limited. Samples in $S_{easy}$ represent prevalent patterns in the data, and focusing on learning from them leads to a lower average expected loss. This enables the model to effectively capture the dominant patterns within the limited dataset size. Moreover, the gradients of the loss function for samples in $S_{easy}$ are smaller, leading to faster convergence and improved model performance within a limited number of training iterations.

*Case 2: Large Dataset* - When the dataset size $|S|$ is large, the model has the capacity to learn complex representations, allowing it to generalize well to both easy and hard samples. As the model learns from samples in both $S_{easy}$ and $S_{hard}$, its overall performance improves, and it achieves higher accuracy on hard samples. Training on samples in $S_{hard}$ helps the model learn more discriminative features, as they often lie close to the decision boundary.

Therefore, in the case of a small dataset, samples with lower LBPE scores are more important.

The use of the LBPE importance metric is outlined in Algorithm 1. LBPE scores over the epochs with the Top-K training accuracy are averaged. The output of this algorithm is the average LBPE score.

### 3.3 Balanced Construction

In this section, we prove that a more balanced class distribution yields a lower expected loss.

**Definition 2.1 (Dataset Selection $S_A$ and $S_B$):** $S_A$ is to select images from each class based on their LBPE scores such that the selection is balanced across classes, and $S_B$ is to select images purely based on their LBPE scores without considering the class balance. Formally, we have:

$$S_A = (x_i, y_i) : x_i \in \mathcal{X}_k, \text{ and } \text{LBPE}_t(x_i, y_i) \leq \tau_k, \qquad S_B = (x_i, y_i) : \text{LBPE}_t(x_i, y_i) \leq \tau \quad (5)$$

where $\mathcal{X}_k$ denotes the set of images from class $k$, and $\tau_k$ is a threshold for class $k$, $\tau$ is a global threshold. Then $S_A$ is a more balanced dataset compared to $S_B$.

**Definition 2.2 (Generalization Error):** The generalization error of a model is the difference between the expected loss on the training dataset and the expected loss on an unseen test dataset:

$$\text{GenErr}(\mathbf{w}) = \mathbb{E}[L_{\text{test}}(\mathbf{w})] - \mathbb{E}[L_{\text{train}}(\mathbf{w})]. \tag{6}$$

**Definition 2.3 (Rademacher Complexity):** The Rademacher complexity [31] of a hypothesis class $\mathcal{H}$ for a dataset $S$ of size $N$ is defined as:

$$\mathcal{R}_N(\mathcal{H}) = \mathbb{E}_{\sigma} \left[ \sup_{h \in \mathcal{H}} \frac{1}{N} \sum_{i=1}^{N} \sigma_i h(\mathbf{x}_i) \right], \tag{7}$$

---

**Algorithm 2** Balanced Dataset Construction

---

**Require:** Condensed dataset $S = (x_i, y_i)_{i=1}^m$ with classes $\mathbf{K}$, LBPE scores $\mathbf{LBPE}$, class-specific thresholds $\boldsymbol{\tau} = \tau_k{}_{k=1}^K$ to ensure an equal number of samples for each class
 1:  Initialize $S_A = \emptyset$                                          ▷ Initialize the balanced subset as an empty set
 2:  **for** each class $k \in \mathbf{K}$ **do**
 3:      $I_{sel} \leftarrow \{i : y_i = k \text{ and } \mathbf{LBPE}_t(x_i, y_i) \leq \tau_k\}$          ▷ Find indices of samples for class $k$
 4:      $S_A \leftarrow S_A \cup (x_i, y_i) : i \in I_{sel}$          ▷ Add the selected samples to the balanced subset
 5:  **end for**
 6:  **return** $S_A$                                          ▷ Return the balanced subset

---

where $\sigma_i$ are independent Rademacher random variables taking values in $-1, 1$ with equal probability.

**Lemma 2.1 (Generalization Error Bound):** With a high probability, the generalization error is upper-bounded by the Rademacher complexity of the hypothesis class:

$$\text{GenErr}(\mathbf{w}) \leq 2\mathcal{R}_N(\mathcal{H}) + \mathcal{O}\left(\frac{1}{\sqrt{N}}\right), \tag{8}$$

where $\mathcal{O}$ represents the order of the term.

**Lemma 2.2 (Rademacher Complexity Comparison):** The Rademacher complexity of dataset $S_A$ is less than that of dataset $S_B$:

$$\mathcal{R}_{N_A}(\mathcal{H}) \leq \mathcal{R}_{N_B}(\mathcal{H}). \tag{9}$$

**Theorem 2.1:** The expected loss for the dataset $S_A$ is less than or equal to $S_B$ when both models achieve similar performance on their respective training sets.

**Proof of Theorem 2.1:** Using Lemma 2.1 and Lemma 2.2, we have:

$$\text{GenErr}(\mathbf{w}_A) \leq \text{GenErr}(\mathbf{w}_B). \tag{10}$$

Assuming that both models achieve similar performance on their respective training sets, the training losses are approximately equal:

$$\mathbb{E}[L_{\text{train}}(\mathbf{w}_A)] \approx \mathbb{E}[L_{\text{train}}(\mathbf{w}_B)]. \tag{11}$$

Given this assumption, we can rewrite the generalization error inequality as:

$$\mathbb{E}[L_{\text{test}}(\mathbf{w}_A)] - \mathbb{E}[L_{\text{train}}(\mathbf{w}_A)] \leq \mathbb{E}[L_{\text{test}}(\mathbf{w}_B)] - \mathbb{E}[L_{\text{train}}(\mathbf{w}_B)]. \tag{12}$$

Adding $\mathbb{E}[L_{\text{train}}(\mathbf{w}_A)]$ to both sides, we get:

$$\mathbb{E}[L_{\text{test}}(\mathbf{w}_A)] \leq \mathbb{E}[L_{\text{test}}(\mathbf{w}_B)]. \tag{13}$$

This result indicates that the balanced dataset $S_A$ is better than $S_B$.

**Theorem 2.2:** Let $S_F$ and $S_C$ be the full and condensed datasets, respectively, and let both $S_F$ and $S_C$ have an imbalanced class distribution with the same degree of imbalance. Then, the influence of the imbalanced class distribution on the expected loss is larger for the condensed dataset $S_C$ than for the full dataset $S_F$.

**Proof of Theorem 2.2:** We compare the expected loss for the full and condensed datasets, taking into account their class imbalances. Let $L(h)$ denote the loss function for the hypothesis $h$. Let $\mathbb{E}[L(h)|S]$ denote the expected loss for the hypothesis $h$ on the dataset $S$. Let $n_{kF}$ and $n_{kC}$ denote the number of samples in class $k$ for datasets $S_F$ and $S_C$, respectively. Let $m_F$ and $m_C$ denote the total number of samples in datasets $S_F$ and $S_C$, respectively. Let $r_k = \frac{n_{kF}}{m_F} = \frac{n_{kC}}{m_C}$ be the class ratio for each class $k$ in both datasets. The expected loss for $S_F$ and $S_C$ can be written as:

$$\mathbb{E}[L(h)|S_F] = \sum_{k=1}^K r_k \mathbb{E}[l(h(x), y)|\mathcal{X}_k], \quad \mathbb{E}[L(h)|S_C] = \sum_{k=1}^K r_k \mathbb{E}[l(h(x), y)|\mathcal{X}_k], \tag{14}$$

To show this, let's compare the expected loss per sample in each dataset:

$$\frac{\mathbb{E}[L(h)|S_C]}{m_C} > \frac{\mathbb{E}[L(h)|S_F]}{m_F}. \tag{15}$$

This implies that the influence of the imbalanced class distribution is larger for $S_C$ than for $S_F$.

**Rule 2: Balanced class distribution should be utilized for the condensed dataset.** The construction of a balanced class distribution based on LBPE scores is outlined in Algorithm 2. Its objective is to create an equal number of samples for each class to ensure a balanced dataset.

# 4 Experiments

## 4.1 Experiment Settings

IPC stands for "Images Per Class". $IPC_{\mathbf{F} \rightarrow \mathbf{T}}$ means flexibly resize the condensed dataset from size $\mathbf{F}$ to size $\mathbf{T}$. More detailed settings can be found in **Appendix** B.1.

**Dataset Condensation Settings.** The CIFAR-10 and CIFAR-100 datasets [20] are condensed via ConvNet-D3 [12], and ImageNet-10 [5] via ResNet10-AP [13], both following IDC [19]. IPC includes 10, 20, or 50, depending on the experiment. For both networks, the learning rate is 0.01 with 0.9 momentum and 0.0005 weight decay. The SGD optimizer and a multi-step learning rate scheduler are used. The training batch size is 64, and the network is trained for $2000 \times 100$ epochs for CIFAR-10/CIFAR-100 and $500 \times 100$ epochs for ImageNet-10.

**YOCO Settings. 1) LBPE score selection.** To reduce computational costs, we derive the LBPE score from training dynamics of early $E$ epochs. To reduce variance, we use the LBPE score from the top-$K$ training epochs with the highest accuracy. For CIFAR-10, we set $E = 100$ and $K = 10$ for all the $IPC_{\mathbf{F}}$ and $IPC_{\mathbf{T}}$. For CIFAR-100 and ImageNet-10, we set $E = 200$ and $K = 10$ for all the $IPC_{\mathbf{F}}$ and $IPC_{\mathbf{T}}$. **2) Balanced construction.** We use $S_A$ in Eq. 5 to achieve a balanced construction. Following IDC [19], we leverage a multi-formation framework to increase the synthetic data quantity while preserving the storage budget. Specifically, an IDC-condensed image is composed of $n^2$ patches. Each patch is derived from one original image with the resolution scaled down by a factor of $1/n^2$. Here, $n$ is referred to as the "factor" in the multi-formation process. For CIFAR-10 and CIFAR-100 datasets, $n = 2$; for ImageNet-10 dataset, $n = 3$. We create balanced classes according to these patches. As a result, all the classes have the same number of samples. **3) Flexible resizing.** For datasets with $IPC_{\mathbf{F}} = 10$ and $IPC_{\mathbf{F}} = 20$, we select $IPC_{\mathbf{T}}$ of $1, 2$, and $5$. For $IPC_{\mathbf{F}} = 50$, we select $IPC_{\mathbf{T}}$ of $1, 2, 5$, and $10$. For a condensed dataset with $IPC_{\mathbf{F}}$, the performance of its flexible resizing is indicated by the average accuracy across different $IPC_{\mathbf{T}}$ values.

**Comparison Baselines.** We have two sets of baselines for comparison: 1) dataset condensation methods including IDC[19], DREAM[24], MTT [1], DSA [50] and KIP [32] and 2) dataset pruning methods including SSP [40], Entropy [3], AUM [35], Forgetting [41], EL2N [34], and CCS [54]. For dataset condensation methods, we use a random subset as the baseline. For dataset pruning methods, their specific metrics are used to rank and prune datasets to the required size.

## 4.2 Primary Results

Tab. 1 provides a comprehensive comparison of different methods for flexibly resizing datasets from an initial $IPC_{\mathbf{F}}$ to a target $IPC_{\mathbf{T}}$. In this table, we have not included ImageNet results on DREAM [24] since it only reports on Tiny ImageNet with a resolution of 64×64, in contrast to ImageNet's 224×224. The third column of the table shows the accuracy of the condensed dataset at the parameter $IPC_{\mathbf{F}}$. We then flexibly resize the dataset from $IPC_{\mathbf{F}}$ to $IPC_{\mathbf{T}}$. The blue area represents the average accuracy across different $IPC_{\mathbf{T}}$ values. For instance, consider the CIFAR-10 dataset with $IPC_{\mathbf{F}} = 10$. Resizing it to $IPC_{\mathbf{F}} = 1, 2$, and $5$ using our method yields accuracies of 42.28%, 46.67%, and 55.96%, respectively. The average accuracy of these three values is 48.30%. This value surpasses the 37.08% accuracy of SSP [40] by a considerable margin of 11.22%.

**Ablation Study.** Tab. 2 shows the ablation study of the LBPE score and the balanced construction across dataset condensation methods. In the first row, the baseline results are shown where neither the LBPE score nor the balanced construction is applied. "Balanced only" (second row) indicates the selection method is random selection and the class distribution is balanced. "LBPE only" (third row) means the

Table 2: Ablation study on two rules. (CIFAR-10: $IPC_{10 \rightarrow 1}$)

| LBPE | Balanced | IDC [19] | DREAM [24] | MTT [1] | KIP [32] |
|:---:|:---:|:---:|:---:|:---:|:---:|
| - | - | 28.23 | 30.87 | 19.75 | 14.06 |
| - | ✓ | 30.19 | 32.83 | 19.09 | 16.27 |
| ✓ | - | 39.38 | 37.30 | 20.37 | 15.78 |
| ✓ | ✓ | **42.28** | **42.29** | **22.02** | **22.24** |

Table 1: IPC means "images per class". Flexibly resize dataset from $IPC_F$ to $IPC_T$ ($IPC_{F \to T}$). The blue areas represent the average accuracy of listed $IPC_T$ datasets for different values of $T$. The gray areas indicate the accuracy difference between the corresponding methods and ours.

| Dataset | $IPC_F$ | Acc. | $IPC_T$ | Condensation | | Pruning Method | | | | | | |
| | | | | IDC[19] | DREAM[24] | SSP[40] | Entropy[3] | AUM[35] | Forg.[41] | EL2N[34] | CCS[54] | Ours |
|---|---|---|---|---|---|---|---|---|---|---|---|---|
| CIFAR-10 | 10 | 67.50 | 1 | 28.23 | 30.87 | 27.83 | 30.30 | 13.30 | 16.68 | 16.95 | 33.54 | **42.28** |
| | | | 2 | 37.10 | 38.88 | 34.95 | 38.88 | 18.44 | 22.13 | 23.26 | 39.20 | **46.67** |
| | | | 5 | 52.92 | 54.23 | 48.47 | 52.85 | 41.40 | 45.49 | 46.58 | 53.23 | **55.96** |
| | | | Avg. | 39.42 | 41.33 | 37.08 | 40.68 | 24.38 | 28.10 | 28.93 | 41.99 | **48.30** |
| | | | Diff. | -8.89 | -6.98 | -11.22 | -7.63 | -23.92 | -20.20 | -19.37 | -6.31 | - |
| | 50 | 74.50 | 1 | 29.45 | 27.61 | 28.99 | 17.95 | 7.21 | 12.23 | 7.95 | 31.28 | **38.77** |
| | | | 2 | 34.27 | 36.11 | 34.51 | 24.46 | 8.67 | 12.17 | 9.47 | 38.71 | **44.54** |
| | | | 5 | 45.85 | 48.28 | 46.38 | 34.12 | 12.85 | 15.55 | 16.03 | 48.19 | **53.04** |
| | | | 10 | 57.71 | 59.11 | 56.81 | 47.61 | 22.92 | 27.01 | 31.33 | 56.80 | **61.10** |
| | | | Avg. | 41.82 | 42.78 | 41.67 | 31.04 | 12.91 | 16.74 | 16.20 | 43.75 | **49.36** |
| | | | Diff. | -7.54 | -6.58 | -7.69 | -18.33 | -36.45 | -32.62 | -33.17 | -5.62 | - |
| CIFAR-100 | 10 | 45.40 | 1 | 14.78 | 15.05 | 14.94 | 11.28 | 3.64 | 6.45 | 5.12 | 18.97 | **22.57** |
| | | | 2 | 22.49 | 21.78 | 20.65 | 16.78 | 5.93 | 10.03 | 8.15 | 25.27 | **29.09** |
| | | | 5 | 34.90 | 35.54 | 30.48 | 29.96 | 17.32 | 21.45 | 22.40 | 36.01 | **38.51** |
| | | | Avg. | 24.06 | 24.12 | 22.02 | 19.34 | 8.96 | 12.64 | 11.89 | 26.75 | **30.06** |
| | | | Diff. | -6.00 | -5.93 | -8.03 | -10.72 | -21.09 | -17.41 | -18.17 | -3.31 | - |
| | 20 | 49.50 | 1 | 13.92 | 13.26 | 14.65 | 5.75 | 2.96 | 7.59 | 4.59 | 18.72 | **23.74** |
| | | | 2 | 20.62 | 20.41 | 20.27 | 8.63 | 3.96 | 10.64 | 6.18 | 24.08 | **29.93** |
| | | | 5 | 31.21 | 31.81 | 30.34 | 17.51 | 8.25 | 17.63 | 11.76 | 32.81 | **38.02** |
| | | | Avg. | 21.92 | 21.83 | 21.75 | 10.63 | 5.06 | 11.95 | 7.51 | 25.20 | **30.56** |
| | | | Diff. | -8.65 | -8.74 | -8.81 | -19.93 | -25.51 | -18.61 | -23.05 | -5.36 | - |
| | 50 | 52.60 | 1 | 13.41 | 13.36 | 15.90 | 1.86 | 2.79 | 9.03 | 4.21 | 19.05 | **23.47** |
| | | | 2 | 20.38 | 19.97 | 21.26 | 2.86 | 3.04 | 12.66 | 5.01 | 24.32 | **29.59** |
| | | | 5 | 29.92 | 29.88 | 29.63 | 6.04 | 4.56 | 20.23 | 7.24 | 31.93 | **37.52** |
| | | | 10 | 37.79 | 37.85 | 36.97 | 13.31 | 8.56 | 29.11 | 11.72 | 38.05 | **42.79** |
| | | | Avg. | 25.38 | 25.27 | 25.94 | 6.02 | 4.74 | 17.76 | 7.05 | 28.34 | **33.34** |
| | | | Diff. | -7.97 | -8.08 | -7.40 | -27.33 | -28.61 | -15.59 | -26.30 | -5.01 | - |
| ImageNet-10 | 10 | 72.80 | 1 | 44.93 | - | 45.69 | 40.98 | 17.84 | 32.07 | 41.00 | 44.27 | **53.91** |
| | | | 2 | 57.84 | - | 58.47 | 52.04 | 29.13 | 44.89 | 54.47 | 56.53 | **59.69** |
| | | | 5 | 67.20 | - | 63.11 | 64.60 | 44.56 | 55.13 | 65.87 | **67.36** | 64.47 |
| | | | Avg. | 56.66 | - | 55.76 | 52.54 | 30.51 | 44.03 | 53.78 | 56.05 | **59.36** |
| | | | Diff. | 2.70 | - | 3.60 | 6.82 | 28.85 | 15.33 | 5.58 | 3.31 | - |
| | 20 | 76.60 | 1 | 42.00 | - | 43.13 | 36.13 | 14.51 | 24.98 | 24.09 | 34.64 | **53.07** |
| | | | 2 | 53.93 | - | 54.82 | 46.91 | 19.09 | 31.27 | 33.16 | 42.22 | **58.96** |
| | | | 5 | 59.56 | - | 61.27 | 56.44 | 27.78 | 36.44 | 46.02 | 57.11 | **64.38** |
| | | | Avg. | 51.83 | - | 53.07 | 46.49 | 20.46 | 30.90 | 34.42 | 44.66 | **58.80** |
| | | | Diff. | 6.97 | - | 5.73 | 12.31 | 38.34 | 27.90 | 24.38 | 14.14 | - |

LBPE score is used for ranking samples, and the selection results are purely based on the LBPE score without considering the class balance. "LBPE + Balanced" indicates that both elements are included for sample selection. The empirical findings conclusively affirm the effectiveness of the two rules, which constitute the principal contributions of our YOCO method.

**Standard Deviation of Experiments.** Different training dynamics and network initializations impact the final results. Therefore, the reported results are averaged over three different training dynamics, and each training dynamic is evaluated based on three different network initializations. See **Appendix** B.2 for the primary results table with standard deviation.

### 4.3 Analysis of Two Rules

#### 4.3.1 Analysis of LBPE Score for Sample Ranking

Tab. 3 illustrates the robust performance of our YOCO method across diverse network structures, including ConvNet, ResNet, and DenseNet, demonstrating its strong generalization ability. Additionally, we present different sample ranking metrics from dataset pruning methods on these networks, demonstrating that our method outperforms both random selection and other data pruning methods.

In Tab. 4, we experiment with prioritizing easy samples over hard ones. We achieve this by reversing the importance metrics introduced by AUM [35], Forg. [41], and EL2N [34] that originally prioritize

Table 3: Accuracies on different network structures and different sample ranking metrics. (IDC [19] condensed CIFAR-10: $IPC_{10 \to 1}$)

|  | ConvNet [12] | ResNet [13] | DenseNet [14] |
|---|---|---|---|
| Random | 28.23 | 24.14 | 24.63 |
| SSP[40] | 27.83 | 24.64 | 24.75 |
| Entropy[3] | 30.30 | 30.53 | 29.93 |
| AUM[35] | 13.30 | 15.04 | 14.56 |
| Forg.[41] | 16.68 | 16.75 | 17.43 |
| EL2N[34] | 16.95 | 19.98 | 21.43 |
| Ours | **42.28** | **34.53** | **34.29** |

Table 4: Prioritizing easy samples is better for different dataset pruning and dataset condensation methods. "$\mathcal{R}?$" represents whether to reverse the metrics which prioritize hard samples. (CIFAR-10: $IPC_{10 \to 1}$)

| Method | $\mathcal{R}?$ | IDC[19] | DREAM[24] | MTT[1] | DSA[50] |
|---|---|---|---|---|---|
| AUM [35] | - | 13.30 | 14.43 | 15.33 | 14.25 |
| AUM [35] | ✓ | 37.97 | 38.18 | 16.63 | 18.23 |
| Forg. [41] | - | 16.68 | 16.26 | 18.82 | 16.55 |
| Forg. [41] | ✓ | 36.69 | 36.15 | 16.65 | 17.03 |
| EL2N [34] | - | 16.95 | 18.13 | 16.98 | 13.14 |
| EL2N [34] | ✓ | 33.11 | 34.36 | 19.01 | 21.29 |
| Ours | - | **42.28** | **42.29** | **22.02** | **22.40** |

Table 5: Balanced construction works on different dataset pruning methods. "$\mathcal{B}?$" represents whether to use balanced construction. The subscript $_{+value}$ indicates the accuracy gain from balanced construction. (IDC [19] condensed CIFAR-10: $IPC_{10 \to T}$)

| $IPC_T$ | $\mathcal{B}?$ | Random | SSP [40] | Entropy [3] | AUM [35] | Forg. [41] | EL2N [34] | Ours |
|---|---|---|---|---|---|---|---|---|
| IPC1 | - | 28.23 | 27.83 | 30.30 | 13.30 | 16.68 | 16.95 | 37.63 |
|  | ✓ | $30.05_{+1.82}$ | $33.21_{+5.38}$ | $33.67_{+3.37}$ | $15.64_{+2.34}$ | $19.09_{+2.41}$ | $18.43_{+1.48}$ | $42.28_{+4.65}$ |
| IPC2 | - | 37.10 | 34.95 | 38.88 | 18.44 | 22.13 | 23.26 | 42.99 |
|  | ✓ | $39.44_{+2.34}$ | $40.57_{+5.62}$ | $42.17_{+3.29}$ | $23.84_{+5.40}$ | $28.06_{+5.93}$ | $26.54_{+3.28}$ | $46.67_{+3.68}$ |
| IPC5 | - | 52.92 | 48.47 | 52.85 | 41.40 | 45.49 | 46.58 | 53.86 |
|  | ✓ | $52.64_{-0.28}$ | $49.44_{+0.97}$ | $54.73_{+1.88}$ | $47.23_{+5.83}$ | $48.02_{+2.53}$ | $48.86_{+2.28}$ | $55.96_{+2.10}$ |

hard samples. Our results indicate that across various condensed datasets, including IDC [19], DREAM [24], MTT [1], and DSA [50], there is a distinct advantage in prioritizing easier samples over harder ones. These findings lend support to our Rule 1.

### 4.3.2  Analysis of Balanced Construction

Fig. 2 presents the class distributions with and without a balanced construction for different datasets and different $IPC_{F \to T}$. As explained in YOCO settings, our balanced construction is based on the multi-formation framework from IDC [19]. Therefore, the x-axis represents the count of images after multi-formation instead of the condensed images. It is evident that a ranking strategy relying solely on the LBPE score can result in a significant class imbalance, particularly severe in the ImageNet dataset. As depicted in Fig. 2(f), three classes have no image patches left. Our balanced construction method effectively mitigates this issue. Notably, in the case of ImageNet-$10_{10 \to 1}$, the balanced construction boosts the accuracy by an impressive 19.37%.

To better understand the impact of balanced class distribution on various dataset pruning methods, we conducted a comparative analysis, as presented in Tab. 5. Clearly, achieving a balanced class distribution significantly enhances the performance of all examined methods. Remarkably, our proposed method consistently outperforms others under both imbalanced and balanced class scenarios, further substantiating the efficacy of our approach.

### 4.4  Other Analysis

**Sample Importance Rules Differ between Condensed Dataset and Full Dataset.** In Fig. 3, we compare Sample Importance Rules for Condensed Datasets ($IPC_{10}$, $IPC_{50}$) and the Full Dataset ($IPC_{5000}$), by adjusting the pruning ratio from 10% to 90%. Unmarked solid lines mean prioritizing easy samples, dashed lines suggest prioritizing hard samples, while marked solid lines depict the accuracy disparity between the preceding accuracies. Therefore, the grey region above zero indicates "Prefer easy samples" ($Rule_{easy}$), while the blue region below zero represents "Prefer hard samples"($Rule_{hard}$). We have two observations. First, as the pruning ratio increases, there is a gradual transition from $Rule_{hard}$ to $Rule_{easy}$. Second, the turning point of this transition depends on the dataset size. Specifically, the turning points for $IPC_{10}$, $IPC_{50}$, and $IPC_{5000}$ occur at pruning

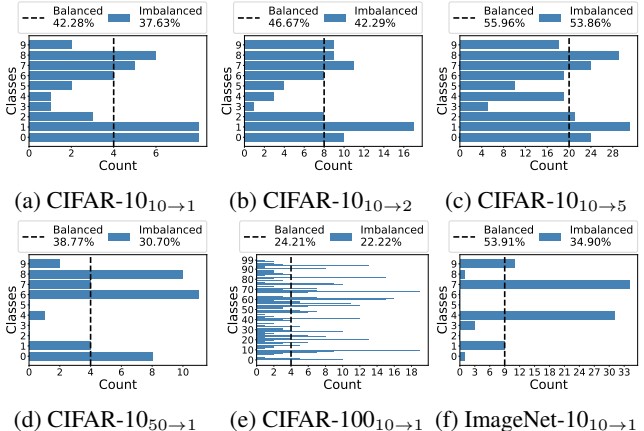

(a) CIFAR-10$_{10 \to 1}$  (b) CIFAR-10$_{10 \to 2}$  (c) CIFAR-10$_{10 \to 5}$

(d) CIFAR-10$_{50 \to 1}$  (e) CIFAR-100$_{10 \to 1}$  (f) ImageNet-10$_{10 \to 1}$

Figure 2: Balanced and imbalanced selection by ranking samples with LBPE score. Dataset$_{\mathbf{F} \to \mathbf{T}}$ denotes resizing the dataset from IPC$_\mathbf{F}$ to IPC$_\mathbf{T}$. Accuracies for each setting are also listed in the legend. (IDC [19] condensed datasets)

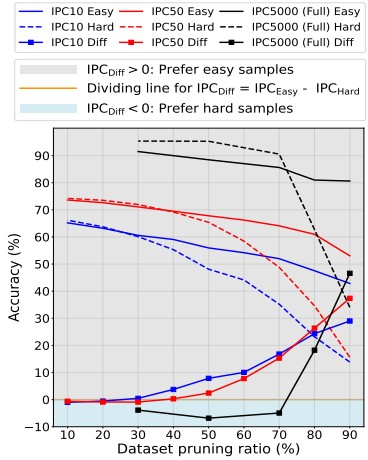

Figure 3: Different sample importance rules between condensed datasets and full datasets.

ratios of 24%, 38%, and 72%, respectively. These experimental outcomes substantiate our Rule 1 that condensed datasets should adhere to Rule$_{easy}$.

**Performance Gap from Multi-formation.** We would like to explain the huge performance gap between multi-formation-based methods (IDC [19] and DREAM [24]) and other methods (MTT [1], KIP [33], and DSA [50]) in Tab. 2 and Tab. 4. The potential reason is that a single image can be decoded to $2^2 = 4$ low-resolution images via multi-formation. As a result, methods employing multi-formation generate four times as many images compared to those that do not use multi-formation. The illustration is shown in Fig. 4.

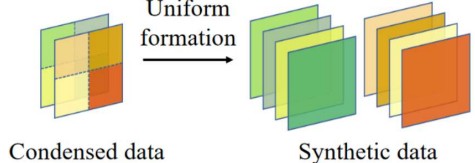

Figure 4: Illustration of the multi-formation with a factor of 2. (Taken from IDC [19])

**Why Use LBPE Score from the Top-$K$ Training Epochs with the Highest Accuracy?** As shown in Eq. 2, different training epoch $t$ leads to a different LBPE score. Fig. 5 illustrates the accuracy of the dataset selected via the LBPE score across specific training epochs. We select LBPE scores from the initial 100 epochs out of 1000 original epochs to reduce computational costs. We have two observations. First, the model's accuracy during the first few epochs is substantially low. LBPE scores derived from these early-stage epochs might not accurately represent the samples' true importance since the model is insufficiently trained. Second, there's significant variance in accuracy even after 40 epochs, leading to potential instability in the LBPE score selection. To address this, we average LBPE scores from epochs with top-$K$ accuracy, thereby reducing variability and ensuring a more reliable sample importance representation.

**Speculating on Why LBPE Score Performs Better at Certain Epochs?** In Fig. 7, we present the distribution of LBPE scores at various training epochs, with scores arranged in ascending order for each class to facilitate comparison across epochs. Our experiment finds the LBPE scores decrease as the epoch number increases. The superior accuracy of LBPE$_{90}$ is due to two reasons. First, the model at the $90_{th}$ epoch is more thoroughly trained than the model at the first epoch, leading to more accurate LBPE scores. Second, the LBPE$_{90}$ score offers a more uniform distribution and a wider range [0, 1], enhancing sample distinction. In contrast, the LBPE$_{1000}$ score is mostly concentrated within a narrow range [0, 0.1] for the majority of classes, limiting differentiation among samples. More possible reasons will be explored in future studies.

**Visualization.** Fig. 6 visualizes the easy and hard samples identified by our YOCO method. We notice that most easy samples have a distinct demarcation between the object and its background. This is particularly evident in the classes of "vacuum cleaner" and "cocktail shaker". The easy samples in

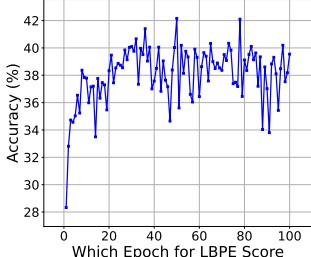

Figure 5: Accuracy of the dataset selected with LBPE score at specific epochs.

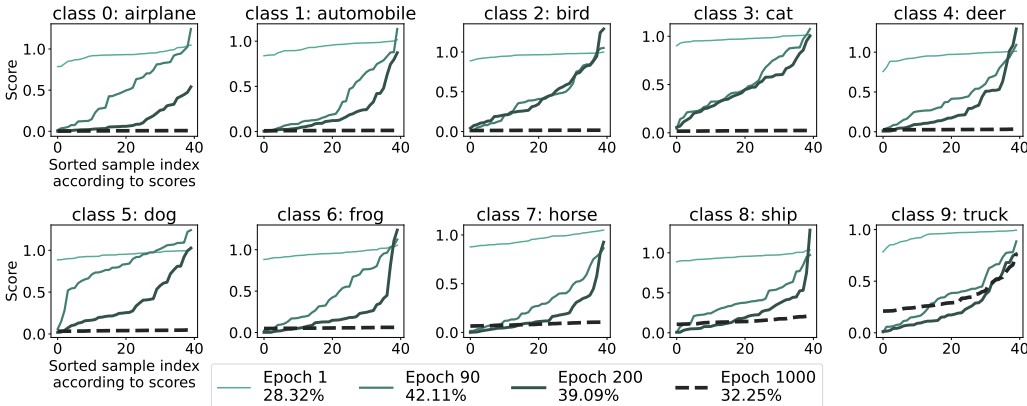

Figure 6: Visualization of hard and easy samples of ImageNet dataset selected by our method. Both the original and condensed images are shown for comparison.

Figure 7: LBPE scores at different epochs (LBPE$_{epoch}$) for ten classes of the CIFAR-10 dataset.

these two classes have clean backgrounds, while the hard samples have complex backgrounds. The visualization provides evidence of our method's ability to identify easy and hard samples.

## 5 Conclusion, Limitation and Future Work

We introduce You Only Condense Once (YOCO), a novel approach that resizes condensed datasets flexibly without an extra condensation process, enabling them to adjust to varying computational constraints. YOCO comprises two key rules. First, YOCO employs the Logit-Based Prediction Error (LBPE) score to rank the importance of training samples and emphasizes the benefit of prioritizing easy samples with low LBPE scores. Second, YOCO underscores the need to address the class imbalance in condensed datasets and utilizes Balanced Construction to solve the problem. Our experiments validated YOCO's effectiveness across different networks and datasets. These insights offer valuable directions for future dataset condensation and dataset pruning research.

We acknowledge several limitations and potential areas for further investigation. First, although our method uses early training epochs to reduce computational costs, determining the sample importance in the first few training epochs or even before training is interesting for future work. Second, we only utilize the LBPE score to establish the importance of samples within the dataset. However, relying on a single metric might not be the optimal approach. There are other importance metrics, such as SSP [40] and AUM [35], that could be beneficial to integrate into our methodology. Third, as our current work only covers clean datasets like CIFAR-10, the performance of our method on noisy datasets requires further investigation.

The border impact is shown in **Appendix** C.

# 6 Acknowledgement

This work is partially supported by Joey Tianyi Zhou's A*STAR SERC Central Research Fund (Use-inspired Basic Research), the Singapore Government's Research, Innovation and enterprise 2020 Plan (Advanced Manufacturing and Engineering domain) under Grant A18A1b0045, and A*STAR CFAR Internship Award for Research Excellence (CIARE). The computational work for this article was partially performed on resources of the National Supercomputing Centre (NSCC), Singapore (`https://www.nscc.sg`).

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
