# Appendix for You Only Condense Once: Two Rules for Pruning Condensed Datasets

**Yang He, Lingao Xiao, Joey Tianyi Zhou**[✉]
CFAR, Agency for Science, Technology and Research, Singapore
IHPC, Agency for Science, Technology and Research, Singapore
{He_Yang, Joey_Zhou}@cfar.a-star.edu.sg

## A  Proof

### A.1  Proof of Lemma 1

**Lemma 1 (Gradient and Importance of Training Samples)**: The gradient of the loss function $\nabla_{\mathbf{w}_t}\mathcal{L}(p(\mathbf{w}_t, x), y)$ for a dataset $S$ is influenced by the samples with prediction errors.

**Proof of Lemma 1**: Let the gradients of the loss function for these two datasets be $\nabla_{\mathbf{w}_t}^{S}\mathcal{L}$ and $\nabla_{\mathbf{w}_t}^{S\neg j}\mathcal{L}$, respectively. Then, we have:

$$\nabla_{\mathbf{w}_t}^{S}\mathcal{L} = \frac{1}{|S|} \sum_{(x,y)\in S} \frac{\partial \mathcal{L}(p(\mathbf{w}_t, x), y)}{\partial p(\mathbf{w}_t, x)} \cdot \frac{\partial p(\mathbf{w}_t, x)}{\partial \mathbf{w}_t}, \tag{1}$$

and

$$\nabla_{\mathbf{w}_t}^{S\neg j}\mathcal{L} = \frac{1}{|S_{\neg j}|} \sum_{(x,y)\in S_{\neg j}} \frac{\partial \mathcal{L}(p(\mathbf{w}_t, x), y)}{\partial p(\mathbf{w}_t, x)} \cdot \frac{\partial p(\mathbf{w}_t, x)}{\partial \mathbf{w}_t}. \tag{2}$$

The difference between the gradients for the two datasets is given by:

$$\Delta\nabla_{\mathbf{w}_t}\mathcal{L} = \frac{1}{|S|} \sum_{(x,y)\in S} \frac{\partial \mathcal{L}(p(\mathbf{w}_t, x), y)}{\partial p(\mathbf{w}_t, x)} \cdot \frac{\partial p(\mathbf{w}_t, x)}{\partial \mathbf{w}_t} - \frac{1}{|S\neg j|} \sum_{(x,y)\in S\neg j} \frac{\partial \mathcal{L}(p(\mathbf{w}_t, x), y)}{\partial p(\mathbf{w}_t, x)} \cdot \frac{\partial p(\mathbf{w}_t, x)}{\partial \mathbf{w}_t}. \tag{3}$$

Split the sum in the first term into two sums, one with the $j$-th sample and one without:

$$\frac{1}{|S|} \left( \sum_{(x,y)\in S_{\neg j}} \frac{\partial \mathcal{L}(p(\mathbf{w}_t, x), y)}{\partial p(\mathbf{w}_t, x)} \cdot \frac{\partial p(\mathbf{w}_t, x)}{\partial \mathbf{w}_t} + \frac{\partial \mathcal{L}(p(\mathbf{w}_t, x_j), y_j)}{\partial p(\mathbf{w}_t, x_j)} \cdot \frac{\partial p(\mathbf{w}_t, x_j)}{\partial \mathbf{w}_t} \right).$$

Notice that the only difference between the sums is the absence of the $j$-th sample in $S_{\neg j}$. Simplify the expression by using $|S_{\neg j}| = |S| - 1$:

$$\Delta\nabla_{\mathbf{w}_t}\mathcal{L} = \left( \frac{1}{|S|} - \frac{1}{|S|-1} \right) \sum_{(x,y)\in S_{\neg j}} \frac{\partial \mathcal{L}(p(\mathbf{w}_t, x), y)}{\partial p(\mathbf{w}_t, x)} \cdot \frac{\partial p(\mathbf{w}_t, x)}{\partial \mathbf{w}_t} + \frac{1}{|S|} \frac{\partial \mathcal{L}(p(\mathbf{w}_t, x_j), y_j)}{\partial p(\mathbf{w}_t, x_j)} \cdot \frac{\partial p(\mathbf{w}_t, x_j)}{\partial \mathbf{w}_t} \tag{4}$$

---

[✉]Corresponding Author

37th Conference on Neural Information Processing Systems (NeurIPS 2023).

Then we have:

$$\Delta \nabla_{\mathbf{w}_t} \mathcal{L} = \frac{-1}{|S|(|S|-1)} \sum_{(x,y)\in S_{\neg j}} \frac{\partial \mathcal{L}(p(\mathbf{w}_t, x), y)}{\partial p(\mathbf{w}_t, x)} \cdot \frac{\partial p(\mathbf{w}_t, x)}{\partial \mathbf{w}_t} + \frac{1}{|S|} \frac{\partial \mathcal{L}(p(\mathbf{w}_t, x_j), y_j)}{\partial p(\mathbf{w}_t, x_j)} \cdot \frac{\partial p(\mathbf{w}_t, x_j)}{\partial \mathbf{w}_t}$$

(5)

## B Experiment

### B.1 Detailed Settings

Our work is mainly based on IDC [4], and we use the open-source code assets for our paper [2] [3]. In section B.1.1 and B.1.2, we will explain the condensation process of IDC. In section B.1.3, we will explain all the methods we compared including dataset condensation and dataset pruning.

#### B.1.1 Dataset and Data Augmentation

**CIFAR-10.** The training set of the original CIFAR-10 dataset [5] contains 10 classes, and each class has 5,000 images with the shape of $32 \times 32$ pixels.

**CIFAR-100.** The training set of the original CIFAR-100 dataset [5] contains 100 classes, and each class contains 500 images with the shape of $32 \times 32$ pixels.

**ImageNet-10.** ImageNet-10 is the subset of ImageNet-1K [3] containing only 10 classes, where each class has on average $1,200$ images of resolution $224 \times 224$.

The augmentations include:

- **Color:** adjusts the brightness, saturation, and contrast of images.
- **Crop:** pads the image and then randomly crops back to the original size.
- **Flip:** flips the images horizontally with a probability of 0.5.
- **Scale:** randomly scale the images by a factor according to a ratio.
- **Rotate:** rotates the image by a random angle according to a ratio.
- **Cutout:** randomly removes square parts of the image, replacing the removed parts with black squares.
- **Mixup:** randomly selects a square region within the image and replaces this region with the corresponding section from another randomly chosen image. It happens at a probability of 0.5.

Following IDC [4], we perform augmentation during training networks in condensation and evaluation, and we use coloring, cropping, flipping, scaling, rotating, and mixup. When updating network parameters, image augmentations are different for each image in a batch; when updating synthetic images, the same augmentations are utilized for the synthetic images and corresponding real images in a batch.

#### B.1.2 Condensation Training and Evaluation

The condensation training process contains three initial components: real images, synthetic images (initially set as a subset of real images), and a randomly initialized network.

**Condensation Training 1: inner loop for 1) image update and 2) network update.** The inner loop of the condensation process involves two parts. The first part is the **update of the synthetic images.** The synthetic images are modified by matching the gradients from the network as it processes both real and synthetic images, with consistent augmentations applied throughout a mini-batch. The second part is the **update of the network.** The network that provides the gradient for images is updated after the image update. During network updates, only real images are used for training. In addition, the network training is limited to an early stage with just 100 epochs for both ConvNet-D3 and ResNet10-AP.

---

[2]https://github.com/snu-mllab/Efficient-Dataset-Condensation
[3]https://github.com/haizhongzheng/Coverage-centric-coreset-selection

**Condensation Training 2: outer loop for network re-initialization.** The outer loop randomly re-initializes the network but does not re-initialize the synthetic images. The outer loop has 2000 and 500 iterations for ConvNet-D3 and ResNet10-AP, respectively.

**Condensation Evaluation.** For condensation evaluation, we need a network trained on the condensed datasets. Unlike condensation training, we fully train the network for 1000 epochs for ConvNet-D3 and ResNet10-AP.

### B.1.3 Details for Other Compared Methods

**Dataset Condensation.** IDC [4] and DREAM [6] use multi-formation of $\text{factor} = \text{n}$, while KIP [7, 8], DSA [13], and MTT [1] contain only one image in a condensed image.

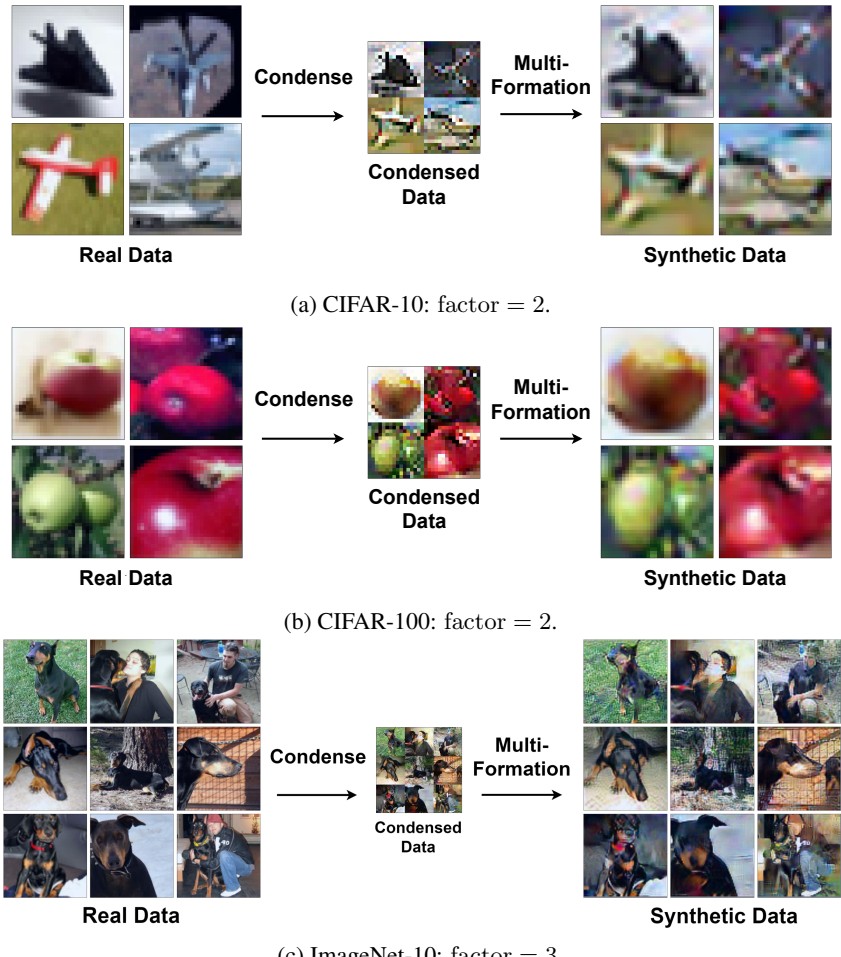

(a) CIFAR-10: $\text{factor} = 2$.

(b) CIFAR-100: $\text{factor} = 2$.

(c) ImageNet-10: $\text{factor} = 3$.

Figure 1: Illustration of multi-formation process.

Fig. 1 illustrates the process of multi-formation (i.e., uniform formation) using a factor $n$. First, real data are down-scaled to $\frac{1}{n^2}$ their size, and $n^2$ number of real data are put together to form condensed data. Condensed data are what we store on disk. During training and evaluation, condensed data undergo a multi-formation process that splits the condensed data into $n^2$ data and restores them to the size of real data.

**Dataset Pruning.** Tab. 1 shows the components required for each dataset pruning method. Based on implementations, dataset pruning baselines can be roughly put into two categories, i.e., 1) model-based and 2) training dynamic-based methods.

Table 1: Components required for each dataset pruning method.

| Method | Model | Training Dynamics | Training Time | Label |
|---|---|---|---|---|
| SSP [11] | ✓ | | Full | |
| Entropy [2] | ✓ | | Full | ✓ |
| AUM [10] | | ✓ | Full | ✓ |
| Forg. [12] | | ✓ | Full | ✓ |
| EL2N [9] | | ✓ | Early | ✓ |
| Ours | | ✓ | Early | ✓ |

1) Model-based methods require a pretrained method for image importance ranking. Self-Supervised Prototype (SSP) [11] utilizes the k-means algorithm to cluster feature spaces extracted from pretrained models. The number of clusters is exactly the number of selected samples, and we select images with the closest distance to the centroid in the feature space. Entropy [2] keeps samples with the largest entropy indicating the maximum uncertainty, and we prune samples with the least entropy. Both methods use ConvNet-D3 models pre-trained with condensed datasets.

2) Training dynamic-based methods keep track of the model training dynamics. AUM [10] keeps hard samples by considering a small area between the correct prediction and the largest logits of other labels. For Forgetting [12], a forgetting event of a sample occurs if the training accuracy of a minibatch containing the sample decreases at the next epoch, and samples with the most forgetting events are considered hard samples. We prune samples with the least forgetting events. If samples contain the same forgetting counts, we prune samples in the order of the index. For EL2N [9], samples with the large norm of error vector are deemed as hard samples, and we prune samples with the least error. The first 10 epochs' training information is averaged to compute the EL2N score. The above methods all choose to prune easy samples when the pruning ratio is small or moderate. For CCS, we use EL2N [9] as the pruning metric, and we prune hard samples with optimal hard cutoff ratios suggested in the paper [14].

## B.2 Main Result with Standard Deviation

Table 2: IPC means "images per class". Flexibly resize dataset from $IPC_F$ to $IPC_T$ ($IPC_{F \to T}$). The subscript $\pm std.$ denotes the standard deviation.

| Dataset | $IPC_F$ | Acc. | $IPC_T$ | Condensation IDC[4] | DREAM[6] | Pruning Method SSP[11] | Entropy[2] | AUM[10] | Forg.[12] | EL2N[9] | CCS[14] | Ours |
|---|---|---|---|---|---|---|---|---|---|---|---|---|
| CIFAR-10 | 10 | 67.50 | 1 | $28.23_{\pm0.08}$ | $30.87_{\pm0.36}$ | $27.83_{\pm0.10}$ | $30.3_{\pm0.27}$ | $13.3_{\pm0.22}$ | $16.68_{\pm0.19}$ | $16.95_{\pm0.12}$ | $33.54_{\pm0.05}$ | $42.28_{\pm0.15}$ |
| | | | 2 | $37.1_{\pm0.34}$ | $38.88_{\pm0.13}$ | $34.95_{\pm0.37}$ | $38.88_{\pm0.03}$ | $18.44_{\pm0.02}$ | $22.13_{\pm0.07}$ | $23.26_{\pm0.25}$ | $39.2_{\pm0.10}$ | $46.67_{\pm0.12}$ |
| | | | 5 | $52.92_{\pm0.06}$ | $54.23_{\pm0.21}$ | $48.47_{\pm0.16}$ | $52.85_{\pm0.11}$ | $41.4_{\pm0.24}$ | $45.49_{\pm0.13}$ | $46.58_{\pm0.34}$ | $53.23_{\pm0.25}$ | $55.96_{\pm0.07}$ |
| | 50 | 74.50 | 1 | $29.45_{\pm0.29}$ | $27.61_{\pm0.15}$ | $28.99_{\pm0.08}$ | $17.95_{\pm0.06}$ | $7.21_{\pm0.08}$ | $12.23_{\pm0.17}$ | $7.95_{\pm0.06}$ | $31.28_{\pm0.21}$ | $38.77_{\pm0.09}$ |
| | | | 2 | $34.27_{\pm0.16}$ | $36.11_{\pm0.27}$ | $34.51_{\pm0.23}$ | $24.46_{\pm0.08}$ | $8.67_{\pm0.17}$ | $12.17_{\pm0.07}$ | $9.47_{\pm0.04}$ | $38.71_{\pm0.25}$ | $44.54_{\pm0.08}$ |
| | | | 5 | $45.85_{\pm0.11}$ | $48.28_{\pm0.14}$ | $46.38_{\pm0.18}$ | $34.12_{\pm0.26}$ | $12.85_{\pm0.08}$ | $15.55_{\pm0.07}$ | $16.03_{\pm0.08}$ | $48.19_{\pm0.16}$ | $53.04_{\pm0.18}$ |
| | | | 10 | $57.71_{\pm0.17}$ | $59.11_{\pm0.06}$ | $56.81_{\pm0.03}$ | $47.61_{\pm0.14}$ | $22.92_{\pm0.11}$ | $27.01_{\pm0.08}$ | $31.33_{\pm0.18}$ | $56.8_{\pm0.11}$ | $61.1_{\pm0.16}$ |
| CIFAR-100 | 10 | 45.40 | 1 | $14.78_{\pm0.10}$ | $15.05_{\pm0.04}$ | $14.94_{\pm0.09}$ | $11.28_{\pm0.07}$ | $3.64_{\pm0.04}$ | $6.45_{\pm0.07}$ | $5.12_{\pm0.06}$ | $18.97_{\pm0.04}$ | $22.57_{\pm0.04}$ |
| | | | 2 | $22.49_{\pm0.12}$ | $21.78_{\pm0.07}$ | $20.65_{\pm0.07}$ | $16.78_{\pm0.03}$ | $5.93_{\pm0.05}$ | $10.03_{\pm0.01}$ | $8.15_{\pm0.04}$ | $25.27_{\pm0.02}$ | $29.09_{\pm0.05}$ |
| | | | 5 | $34.9_{\pm0.02}$ | $35.54_{\pm0.04}$ | $30.48_{\pm0.04}$ | $29.96_{\pm0.12}$ | $17.32_{\pm0.10}$ | $21.45_{\pm0.14}$ | $22.4_{\pm0.09}$ | $36.01_{\pm0.05}$ | $38.51_{\pm0.05}$ |
| | 20 | 49.50 | 1 | $13.92_{\pm0.03}$ | $13.26_{\pm0.04}$ | $14.65_{\pm0.05}$ | $5.75_{\pm0.02}$ | $2.96_{\pm0.01}$ | $7.59_{\pm0.06}$ | $4.59_{\pm0.05}$ | $18.72_{\pm0.11}$ | $23.74_{\pm0.19}$ |
| | | | 2 | $20.62_{\pm0.03}$ | $20.41_{\pm0.08}$ | $20.27_{\pm0.09}$ | $8.63_{\pm0.02}$ | $3.96_{\pm0.04}$ | $10.64_{\pm0.07}$ | $6.18_{\pm0.04}$ | $24.08_{\pm0.01}$ | $29.93_{\pm0.13}$ |
| | | | 5 | $31.21_{\pm0.09}$ | $31.81_{\pm0.03}$ | $30.34_{\pm0.10}$ | $17.51_{\pm0.08}$ | $8.25_{\pm0.07}$ | $17.63_{\pm0.04}$ | $11.76_{\pm0.13}$ | $32.81_{\pm0.11}$ | $38.02_{\pm0.08}$ |
| | 50 | 52.60 | 1 | $13.41_{\pm0.02}$ | $13.36_{\pm0.01}$ | $15.9_{\pm0.10}$ | $1.86_{\pm0.03}$ | $2.79_{\pm0.03}$ | $9.03_{\pm0.08}$ | $4.21_{\pm0.04}$ | $19.05_{\pm0.04}$ | $23.47_{\pm0.11}$ |
| | | | 2 | $20.38_{\pm0.11}$ | $19.97_{\pm0.21}$ | $21.26_{\pm0.15}$ | $2.86_{\pm0.05}$ | $3.04_{\pm0.02}$ | $12.66_{\pm0.11}$ | $5.01_{\pm0.05}$ | $24.32_{\pm0.07}$ | $29.59_{\pm0.11}$ |
| | | | 5 | $29.92_{\pm0.07}$ | $29.88_{\pm0.09}$ | $29.63_{\pm0.22}$ | $6.04_{\pm0.05}$ | $4.56_{\pm0.10}$ | $20.23_{\pm0.07}$ | $7.24_{\pm0.11}$ | $31.93_{\pm0.06}$ | $37.52_{\pm0.00}$ |
| | | | 10 | $37.79_{\pm0.01}$ | $37.85_{\pm0.12}$ | $36.97_{\pm0.07}$ | $13.31_{\pm0.10}$ | $8.56_{\pm0.08}$ | $29.11_{\pm0.08}$ | $11.72_{\pm0.06}$ | $38.05_{\pm0.09}$ | $42.79_{\pm0.06}$ |
| ImageNet Subset-10 | 10 | 72.80 | 1 | $44.93_{\pm0.37}$ | - | $45.69_{\pm0.74}$ | $40.98_{\pm0.39}$ | $17.84_{\pm0.45}$ | $32.07_{\pm0.17}$ | $41.0_{\pm0.71}$ | $44.27_{\pm0.80}$ | $53.91_{\pm0.49}$ |
| | | | 2 | $57.84_{\pm0.10}$ | - | $58.47_{\pm0.42}$ | $52.04_{\pm0.29}$ | $29.13_{\pm0.25}$ | $44.89_{\pm0.41}$ | $54.47_{\pm0.40}$ | $56.53_{\pm0.13}$ | $59.69_{\pm0.19}$ |
| | | | 5 | $67.2_{\pm0.52}$ | - | $63.11_{\pm0.63}$ | $64.6_{\pm0.09}$ | $44.56_{\pm0.66}$ | $55.13_{\pm0.19}$ | $65.87_{\pm0.04}$ | $67.36_{\pm0.35}$ | $64.47_{\pm0.36}$ |
| | 20 | 76.60 | 1 | $42.0_{\pm0.28}$ | - | $43.13_{\pm0.5}$ | $36.13_{\pm0.17}$ | $14.51_{\pm0.12}$ | $24.98_{\pm0.13}$ | $24.09_{\pm0.67}$ | $34.64_{\pm0.16}$ | $53.07_{\pm0.31}$ |
| | | | 2 | $53.93_{\pm0.31}$ | - | $54.82_{\pm0.27}$ | $46.91_{\pm0.51}$ | $19.09_{\pm0.45}$ | $31.27_{\pm0.23}$ | $33.16_{\pm0.41}$ | $42.22_{\pm0.17}$ | $58.96_{\pm0.36}$ |
| | | | 5 | $59.56_{\pm0.29}$ | - | $61.27_{\pm0.67}$ | $56.44_{\pm0.95}$ | $27.78_{\pm0.13}$ | $36.44_{\pm0.53}$ | $46.02_{\pm0.26}$ | $57.11_{\pm0.2}$ | $64.38_{\pm0.38}$ |

Tab. 2 provides the standard deviations for our main result. We note that standard deviations for random selection are averaged over three random score selections and for each set of randomly selected images, the accuracy is averaged over three randomly initialized networks. For dataset

pruning methods, the reported results are averaged over three different training dynamics, and each training dynamic is evaluated based on three different network initializations.

## C  Broader Impact

The broader impact of our research lies in its potential to revolutionize the efficiency of deep learning models. By allowing for the flexible resizing of condensed datasets, our method can enable these models to deliver optimal performance under varying computational constraints. This could lead to more efficient use of resources, potentially making deep learning more accessible for devices with limited computational power. As a result, our work holds significant promise in driving forward the field of deep learning, particularly in the context of dataset efficiency.

The fact that our method has been validated only on clean datasets like CIFAR-10, CIFAR-100, and ImageNet raises questions about its robustness in real-world scenarios where data is often noisy and imperfect. Real-world datasets typically contain inconsistencies, outliers, or errors, which may challenge our method's ability to prioritize samples based on the LBPE score accurately. Such datasets may introduce more complex patterns that could potentially be characterized as "hard" samples, and these might be overlooked by our current method. Hence, it is essential to explore how our YOCO method performs under such conditions. It's also important to investigate possible modifications or enhancements to our method that could improve its robustness to noise and variability, such as integrating additional metrics for sample importance or developing methods to handle noisy data effectively. This broader impact highlights the ongoing need for research that not only strives for efficiency and adaptability in deep learning models but also robustness and resilience in real-world scenarios.