# OpenReview forum: "You Only Condense Once: Two Rules for Pruning Condensed Datasets"
_NeurIPS.cc/2023/Conference — NeurIPS 2023 poster_

### Official Review · Reviewer_hpsG · 2023-07-01

**Soundness:** 4 excellent
**Presentation:** 4 excellent
**Contribution:** 3 good
**Rating:** 7
**Confidence:** 5

**Summary:**

This paper proposes a combination method for dataset pruning and dataset condensation. Both methods could reduce the size of training datasets. Experiments show that the proposed method achieves state-of-the-art performance on several datasets regardless of networks. The proposed rules are simple and effective, which might be commonly used in future works.

**Strengths:**

- This paper studies an interesting yet rarely studied problem, which is important for on-device scenarios. In addition, the authors have comprehensively analyze the challenges of this problem and the drawback of existing methods.

- This paper is well-written and easy to follow. I generally enjoy reading the story, motivation, method and experiments of this paper.

- Two novel, simple, and effective rules/metrics are introduced to select informative and balanced samples. Importantly, with this two rules, one can carry on various size requirements by one process.

- Extensive experiments, including results and visualization, demonstrate the effectiveness and efficiency of the proposed method, and show the superiority over the state-of-the-art methods.


**Weaknesses:**

Generally, I think the paper is strong. However, I have the following questions.

- In Table 1, the ImageNet-10 on the DREAM method is not listed.

- Some proofs, such as Proof of Theorem 1, can be shortened.

- The core idea is first to condense the dataset before pruning the dataset. How about pruning the dataset first and then condensing it? Which way is better and why?

**Questions:**

- From Figure 1, previous methods require 200K *(N-1) epochs for extra condensation. Is there any way to reduce the extra condensation process without using dataset pruning? Dataset Pruning is just like a selection method.

- Will the code be open-sourced?

**Limitations:**

Yes.

---

> ### Author Rebuttal · Authors · 2023-08-05
>
> Thanks a lot for your comment. We will answer the questions below.
>
> >1. In Table 1, the ImageNet-10 on the DREAM method is not listed.
>
> DREAM uses tiny-ImageNet (resolution: 64x64) instead of a subset of ImageNet (resolution: 224 x 224). This is because DREAM performs clustering during condensation, and the increase in the image resolution incurs large memory costs.
>
>  > 2. Some proofs, such as Proof of Theorem 1, can be shortened.
>
> Thanks for the suggestion. We will modify this part.
>
> > 3  The core idea is first to condense the dataset before pruning the dataset. How about pruning the dataset first and then condensing it? Which way is better and why?
>
> Thanks for the insightful question. We believe conducting dataset pruning before dataset condensation is better.
> Dataset pruning has roughly two times compression, and dataset condensation has 100 times compression. If we do dataset pruning before dataset condensation, the final compression ratio is still 100 times. If we do dataset condensation before dataset pruning, the final compression ratio can reach 200 times.
>
> > 4  From Figure 1, previous methods require 200K\*(N-1) epochs for extra condensation. Is there any way to reduce the extra condensation process without using dataset pruning? Dataset Pruning is just like a selection method.
>
> Yes, it is possible. Thanks for your suggestion.
>
> >  5. Will the code be open-sourced?
>
> Yes. The code will be released after acceptance.

---

> > ### Comment · Reviewer_hpsG · 2023-08-12
> > **Response to rebuttal**
> >
> > Thanks for your effort in rebuttal. My concerns have been well addressed in the response.

---

### Official Review · Reviewer_hLEg · 2023-07-04

**Soundness:** 3 good
**Presentation:** 3 good
**Contribution:** 4 excellent
**Rating:** 8
**Confidence:** 4

**Summary:**

This paper target at the dataset condensation for on-device scenarios.
It proposes two novel rules to prune the condensed dataset. The first rule is to rank the condensed images. The rule believes easy examples are better for pruning condensed dataset. The second rule is to choose balanced samples for different classes. The overall writing is clear. The experiments are sufficient.


**Strengths:**

1. This is the first work to consider pruning the condensed dataset. The novelty is good.
2. The illustration and writing are clear. The motivation is clearly explained in the figures.
3. The paper provides solid proof for the LBPE metric for ranking the condensed images.
4. The balanced construction is reasonable expecially when the dataset size is small.
5. The experiment looks good. On CIFAR-10, CIFAR-100, and ImageNet, the proposed YOCO method surpasses other methods by a large margin. For example, on CIFAR-10, the improvements are from 6.57% to 24.33%. On CIFAR-100, the improvements are from 4.77% to 22.13%.
6. The analysis is interesting. It's interesting to see that the rules are different between the condensed dataset and the full dataset.

**Weaknesses:**

1. The LBPE score and balanced construction. Which rule is more important?
2. If we view the condensed dataset as a full dataset, will the rules change? Is there any threshold for when the rules will change?
3. The LBPE Score is from the Top-K Training Epochs with the Highest Accuracy. Why not use the loss to extract the LBPE score?

**Questions:**

Please see the Weaknesses.

**Limitations:**

The limitation part is covered in section 5.

---

> ### Author Rebuttal · Authors · 2023-08-05
>
> Thank you for your high score, and we'd like to answer your question below.
>
> > 1. The LBPE score and balanced construction. Which rule is more important?
>
> Thanks for the question. Table 2 in the paper provides an ablation study on the importance of LBPE score and Balanced Construction. These results indicate a higher importance of LBPE score over balanced construction.
>
> > 2.  If we view the condensed dataset as a full dataset, will the rules change? Is there any threshold for when the rules will change?
>
> Thanks for the question.  In our setting, we indeed view the "condensed dataset" IPCF as the "full dataset". As reported in line 261-262, the threshold is 24%, 38%, and 72% of total data pruned for condensed CIFAR10 IPCF=10, IPCF=50, and full CIFAR10 dataset, respectively. The threshold tends to occur at a large pruning ratio if the dataset is large.
>
> > 3.  The LBPE Score is from the Top-K Training Epochs with the Highest Accuracy. Why not use the loss to extract the LBPE score?
>
> Thanks for the question. We agree that the loss could be used to extract the LBPE score. In the following two tables, we compare the performance for using "accuracy" and "loss" as metrics.
>
> Table 1. CIFAR10 IPCF=10
>
> | (IPC10)  | ipc1  | ipc2  | ipc5  | average |
> | -------- | ----- | ----- | ----- | ------- |
> | accuracy | 42.79 | 46.28 | 55.29 | 48.12   |
> | loss     | 41.42 | 47.12 | 55.96 | 48.16   |
>
>
> Table 2. CIFAR10 IPCF=50
>
> | (IPC50)  | ipc1  | ipc2  | ipc5  | ipc10 | average |
> | -------- | ----- | ----- | ----- | ----- | ------- |
> | accuracy | 39.88 | 45.68 | 53.28 | 60.31 | 49.79   |
> | loss     | 37.01 | 45.47 | 53.86 | 60.96 | 48.16   |
>
> The results show that these two metrics achieve similar performance on small IPCF, but  "accuracy" metric is better on large IPCF.

---

> > ### Comment · Reviewer_hLEg · 2023-08-21
> > **------------------Post rebuttal-----------**
> >
> > Thanks for well addressing my concern. I maintain my scoring.

---

### Official Review · Reviewer_9GkH · 2023-07-06

**Soundness:** 2 fair
**Presentation:** 2 fair
**Contribution:** 3 good
**Rating:** 6
**Confidence:** 4

**Summary:**

This paper employs a dataset pruning approach on a condensed dataset. Specifically, the intent is to evaluate and rank the data within this condensed dataset based on their respective significance. This enables users to selectively choose any number of condensed datasets, offering both efficacy and flexibility in training the model.

**Strengths:**

This paper claims that it's the first work to conduct dataset pruning on condensed datasets to fit varying computational constraints.
LBPE score is a reasonable metric to measure the importance of data in the condensed dataset.


**Weaknesses:**

1. The main weakness is the evaluation part. This paper compares their method: (1). dataset condensation + dataset pruning with the previous method (2). dataset condensation + random selection (line 202), which is not fair. For example, it's better to show the effectiveness of the proposed method by comparing (1) with IPCF = 10 to IPCT=1 or 2 or 5 with (3): directly generating the dataset condensation with the IPC=1 or 2 or 5. If it can get similar performance, this paper can claim its advantages of training time.

2. It's unclear—the whole pipeline in Fig 1 or the rest of the paper. I can only understand the paper after reading the Appendix. It's more critical than putting the algorithm part in the main body.

3. Proof of Theorem 1 is not solid.

4. The imbalanced part is not thoroughly analyzed. The equal number of each class is a standard setting in the condensed dataset. This paper does not provide new insight from Section 3.3

**Questions:**

Please refer to Weakness.

---

> ### Author Rebuttal · Authors · 2023-08-04
>
> Thank you for your comments regarding the evaluation section of our paper. We will explain your questions point by point.
>
> > 1. The main weakness is the evaluation part. This paper compares their method: (1). dataset condensation + dataset pruning with the previous method (2). dataset condensation + random selection (line 202), which is not fair. For example, it's better to show the effectiveness of the proposed method by comparing (1) with IPCF = 10 to IPCT=1 or 2 or 5 with (3): directly generating the dataset condensation with the IPC=1 or 2 or 5. If it can get similar performance, this paper can claim its advantages of training time.
>
> Thanks for your question. We will explain our comparison is fair in two parts.
>
> ### Comparison with dataset pruning method
> We would like to clarify that this paper, as the title suggests, utilizes **dataset pruning** techniques to fit condensed datasets on various computational budgets. So we compare the results with **dataset pruning** methods, including SSP [1],  Entropy [2], AUM [3], Forg. [4], and EL2N [5] in **Table 1** and **Table 5**. It looks like the reviewer ignored these comparisons. We want to emphasize that these comparisons are fair.
>
> ### Comparison with dataset condensation method
> Regarding the comparison with dataset condensation methods, we believe it is also fair. In the table below, we list the required time for the condensation process on GPU 3090. We have these observations below:
>
> A. Condensation on IPC-10 requires 12 hrs 36 mins.
>
> B. Our method requires 12 hrs 37 mins. It consists of the 12 hrs 36 mins required by process A, and **only 1 minute required by our pruning method**.
>
> Following the suggestion from the reviewer that we should not use random selection as a baseline, we set the accuracies of other IPCs to zeros. We also include average accuracy over IPC from 1 to 10 as the metric. It is clear that our method improve the average accuracy by **+50.03%** (from 6.75% to 56.78%) with almost **the same training time** required compared to "condense ipc10 only".
>
>
> |                        | ipc1  | ipc2  | ipc3 | ipc4  | ipc5  | ipc6  | ipc7 | ipc8  | ipc9  | ipc10 | average | time            |   |
> |------------------------|-------|-------|------|-------|-------|-------|------|-------|-------|-------|---------|-----------------|---|
> | condense ipc1 only     | 50.8  | 0     | 0    | 0     | 0     | 0     | 0    | 0     | 0     | 0     | 5.08    | 11 hrs 46 mins  |   |
> | condense ipc2 only     | 0     | 54.8  | 0    | 0     | 0     | 0     | 0    | 0     | 0     | 0     | 5.48    | 11 hrs 48 mins  |   |
> | condense ipc3 only     | 0     | 0     | 59.8 | 0     | 0     | 0     | 0    | 0     | 0     | 0     | 5.98    | 11 hrs 50 mins  |   |
> | condense ipc4 only     | 0     | 0     | 0    | 61.8  | 0     | 0     | 0    | 0     | 0     | 0     | 6.18    | 11 hrs 51 mins  |   |
> | condense ipc5 only     | 0     | 0     | 0    | 0     | 62.5  | 0     | 0    | 0     | 0     | 0     | 6.25    | 11 hrs 53 mins  |   |
> | condense ipc6 only     | 0     | 0     | 0    | 0     | 0     | 64.6  | 0    | 0     | 0     | 0     | 6.46    | 12 hrs 02 mins  |   |
> | condense ipc7 only     | 0     | 0     | 0    | 0     | 0     | 0     | 65.5 | 0     | 0     | 0     | 6.55    | 12 hrs 19 mins  |   |
> | condense ipc8 only     | 0     | 0     | 0    | 0     | 0     | 0     | 0    | 66.3  | 0     | 0     | 6.63    | 12 hrs 23 mins  |   |
> | condense ipc9 only     | 0     | 0     | 0    | 0     | 0     | 0     | 0    | 0     | 66.8  | 0     | 6.68    | 12 hrs 33 mins  |   |
> | condense ipc10 only    | 0     | 0     | 0    | 0     | 0     | 0     | 0    | 0     | 0     | 67.5  | 6.75    | 12 hrs 36 mins  |   |
> | **ours**                   | 42.79 | 46.28 | 51.8 | 54.27 | 55.29 | 59.61 | 61.1 | 63.68 | 65.44 | 67.5  | **56.78**   | 12 hrs 37 mins  |   |
>
>
>
> ### Reference
>
> [1] Beyond neural scaling laws: beating power law scaling via data pruning, NeurIPS 2022
>
> [2] Selection via Proxy: Efficient Data Selection for Deep Learning, ICLR 2020
>
> [3] Identifying Mislabeled Data using the Area Under the Margin Ranking, NeurIPS 2020
>
> [4] An Empirical Study of Example Forgetting during Deep Neural Network Learning, ICLR 2019
>
> [5] Deep Learning on a Data Diet: Finding Important Examples Early in Training, NeurIPS 2021
>
>
> > 2. It's unclear—the whole pipeline in Fig 1 or the rest of the paper. I can only understand the paper after reading the Appendix. It's more critical than putting the algorithm part in the main body.
>
> Thanks for your question. But we are sorry that we don't quite understand this question.
> **Concerning Figure 1**: While we recognize that the reviewer found this figure unclear, we are unsure which part is unclear. As commented by Reviewer hLEg, "The motivation is clearly explained in the figures."
>
>
> > 3. Proof of Theorem 1 is not solid.
>
> Thanks for your question. But it's difficult for us to provide a response based on the general judgment of "not solid". Could you please provide more information?
>
>
> > 4. The imbalanced part is not thoroughly analyzed. The equal number of each class is a standard setting in the condensed dataset. This paper does not provide new insight from Section 3.3
>
> Thanks for the clear question. Regarding the " imbalanced part", please take a look at **Section 4.3.2.**, **Table 3**, and **Figure 5**. We use these to analyze the imbalanced part. Could the reviewer please provide more information on which aspect of the analysis is missing for the imbalanced part?
>
> Regarding your comment "The equal number of each class is a standard setting in the condensed dataset", we agree with this. However, the equal number of each class is **NOT** a standard setting in **dataset pruning** methods. We find that the equal number of each class is quite useful for **pruning** condensed dataset. This is our new insight, and we sincerely hope the reviewer will recognize this.

---

> > ### Comment · Reviewer_9GkH · 2023-08-10
> > **Thank you for the clarification**
> >
> > I appreciate the effort put forth in this paper and recognize the novelty in exploring dataset pruning, a direction that I believe remains relatively under-researched. I do have several points of contention and suggestions that I would like to highlight:
> >
> > Q1. The main concern still exists. Referring to Table 1, the paper demonstrates that using IDC directly to produce IPC=10 results in an accuracy of 67.5%. However, when the proposed dataset pruning is employed on a condensed dataset with IPC=10 from IPC=50, the accuracy drops to 60.31%. Such a significant decrease in accuracy challenges the practicality of this pruning method. The primary objective of pruning is to derive a compact model (dataset in this paper) with only a slight compromise on accuracy when compared to a pre-trained model. And this pruned model should outperform training from scratch. While I acknowledge that pruning can speed up the process when leveraging a pre-trained dataset, this significant drop in accuracy remains concerning. My central point here is, I'd prefer comparisons between the Pruned Condensed Dataset and directly obtaining a Small Condensed Dataset with the same IPC. If the latter yields better results without requiring the intermediate step of dataset pruning, then what merits does the pruning method offer?
> >
> > Q2. I do not have any major concerns here. This isn't a critical issue but more of a suggestion for improvement.
> >
> > Q3. The proof provided for Theorem 1 doesn't strike me as formal proof; it seems more like an elucidation. I'd recommend refining this section to meet the standards of rigorous proof.
> >
> > Q4. I acknowledge and commend the analysis of the imbalanced part. My primary issue lies in "The equal number of each class is a standard setting in the condensed dataset." this is not a new insight. In my understanding, the primary aim of dataset pruning is to extract a succinct yet informative condensed dataset. So the goal is to compare with the methods of the condensed dataset.  Previous methods of condensed datasets seem to consistently adopt the Balanced construction. For this point, there is not a technical concern but only a novelty concern.
> >
> > In summary, I do acknowledge and commend the contributions of 1, 2, and partial 3 in the paper. However, because this paper is to focus on the first work using dataset pruning on the condensed dataset, authors should emphasize the effectiveness of dataset pruning in this field, such as litter accuracy drop from pre-trained condensed dataset or better than generated condensed dataset directly generated from scratch (IPC, DREAM), instead of focusing on comparing with other unproposed dataset pruning condensed dataset method. Regarding the experimental results, It would be more compelling to demonstrate superior performance when using the Pruned Large Pre-trained Condensed Dataset compared to directly generating the Small Condensed Dataset persist.

---

> > > ### Author Response · Authors · 2023-08-11
> > > **Response to Q1 (the main concern)**
> > >
> > > > Regarding the experimental results, It would be more compelling to demonstrate superior performance when using the Pruned Large Pre-trained Condensed Dataset compared to directly generating the Small Condensed Dataset persist.
> > >
> > > > Referring to Table 1, the paper demonstrates that using IDC directly to produce IPC=10 results in an accuracy of 67.5%. However, when the proposed dataset pruning is employed on a condensed dataset with IPC=10 from IPC=50, the accuracy drops to 60.31%. Such a significant decrease in accuracy challenges the practicality of this pruning method.
> > >
> > > Thank you very much for your reply. We totally agree that the mentioned **superior performance** is a very good direction to explore. However, what we want to present in this paper is **increasing flexibility** for already condensed datasets.
> > >
> > > Let us look into the example the reviewer gave. We add setting C for comparison, and analysis point by point.
> > >
> > > **A**. IDC directly produces IPC=10 results, and the accuracy is **67.5%**.
> > >
> > > **B**. The proposed dataset pruning method selects IPC=10 from IPC=50, and the accuracy is **60.31%**.
> > >
> > > **C**. Based on IPC-10, our pruning method improves the accuracy of IPC 1, 2, 3,...,9.  At the same time, IPC-10 still holds the accuracy of **67.5%**. The extra time required is just **one minute**.
> > >
> > >
> > > The analysis of setting A, B and C is listed below:
> > >
> > > 1. We admit that on a **single** IPC-10, setting A is much better than setting B. But please consider the scenario that we need **multiple** IPC for on-device applications. Therefore, please compare setting A with setting C: our method holds the 67.5% accuracy of the IPC-10, and improves all the accuracy of IPC 1,2,...9. All the improvements are achieved with **one extra minute**.
> > >
> > > 2. We hope you don't mind us saying this, but setting B you brought up is a little unfair. This is because we can select IPC-10 from **any larger IPC-X** including IPC-20, IPC-50, IPC-100, IPC-200, IPC-500 and so on. We find that if the number of IPC-X **increases**, the performance of the selected IPC-10 will **decrease**. This found pattern can be understood by the increasing pruning ratio. For example, selecting IPC-10 from IPC-20 leads to pruning ratio = 50%. And selecting IPC-10 from IPC-50 leads to pruning ratio = 80%. Considering the pruning ratio is 80%, it means that **80% of the information from the original dataset is abandoned**. It is very difficult to achieve comparable performance to a “directly condensed IPC-10”, which has the full dataset information.
> > >
> > > 3. Following point 2, the pruning ratio defined in setting B is actually **80%** (IPC-10 from IPC-50). We want to share that when pruning **80%** images from **full** CIFAR-10 dataset, the state-of-the-art pruning method [1] still has an accuracy drop of **4.30%** (from 95.23% to 90.93%). These numbers are shown in Table 2 and Figure 5 (a) in [1]. Please note this 4.30% accuracy drop is for **full** **dataset**, and pruning **condensed** **dataset** is much more difficult. Therefore, we believe our **7.19%** accuracy drop (from 67.5% to 60.31%) is a very good result.
> > >
> > > 4. The table below shows the accuracy of selecting IPC-10 from IPC-50. We totally understand the reviewer wants to compare **column to column** using 67.5% with 60.31%. This comparison is actually **unfair** since their pruning ratio is different (0% vs 80%). Our achievement is obtained by comparing **row to row** using 60.31% with 54.72%. We believe it is **fair** because these two numbers (60.31% with 54.72%) are obtained under the same pruning ratio (80%). Our pruning method beats random selection by a large margin.
> > >
> > >
> > > |                         | IPC10 (baseline) | Select IPC10 from IPC50 |
> > > | ----------------------- | ---------------- | ----------------------- |
> > > | Pruning ratio (\%)      | 0                | 80\%                    |
> > > | Random selection (\%)   | 67.50            | 54.72                   |
> > > | Our pruning method (\%) | 67.50            | **60.31**               |
> > >
> > >
> > >
> > > In conclusion, for any given IPC-X, using our pruning method will produce better accuracy for IPC-1, IPC-2, IPC-3,... IPC-(X-1). At the same time, we maintain the accuracy of IPC-X. We sincerely hope the reviewer can recognize our contribution regarding **increasing flexibility**.
> > >
> > > We thank you again for the valuable discussion brought by the reviewer. We will incorporate these discussions to revise our submission and make it more clear. Thank you!
> > >
> > > ## Reference
> > >
> > > [1] Zheng, Haizhong, et al. "Coverage-centric Coreset Selection for High Pruning Rates." in ICLR, 2023.

---

> > > ### Author Response · Authors · 2023-08-11
> > > **Response to Q2, Q3, and Q4**
> > >
> > > > Q2. I do not have any major concerns here. This isn't a critical issue but more of a suggestion for improvement.
> > >
> > > > Q3. The proof provided for Theorem 1 doesn't strike me as formal proof; it seems more like an elucidation. I'd recommend refining this section to meet the standards of rigorous proof.
> > >
> > > Thank you for your insightful suggestions. We will follow these suggestions to improve our paper.
> > >
> > > > Q4. I acknowledge and commend the analysis of the imbalanced part. My primary issue lies in "The equal number of each class is a standard setting in the condensed dataset." this is not a new insight. In my understanding, the primary aim of dataset pruning is to extract a succinct yet informative condensed dataset. So the goal is to compare with the methods of the condensed dataset. Previous methods of condensed datasets seem to consistently adopt the Balanced construction. For this point, there is not a technical concern but only a novelty concern.
> > >
> > > Thank you for the comments.
> > >
> > > First, please find the table below to conclude the previous literature. Dataset Condensation methods use **balanced** construction, while Dataset Pruning methods [1,2,3,4,5] use **imbalanced** construction.
> > >
> > > | Previous Literature | Balance or Imbalance |
> > > | ----------------------- | ----- |
> > > | Dataset Condensation  | Balance |
> > > | Dataset Pruning  | Imbalance |
> > >
> > > Second, please note that our target is to do dataset **condensation** first, and do dataset **pruning** after. The table below lists two possible ways to achieve our target.
> > >
> > > |                      | Direct Combination | Ours Contribution |
> > > | -------------------- | ------------------ | ----------------- |
> > > | Dataset Condensation | Balance            | Balance           |
> > > | Dataset Pruning      | Imbalance          | **Balance**           |
> > >
> > > - "Direct Combination" is a direct combination of two previous methods to achieve our target. This combination provides no novel contribution.
> > > - "Ours Contribution" shows we are the **first** to use **balanced** construction for pruning condensed dataset.
> > >
> > > In conclusion, our novelty is that we are the **first** to discover the pruning rules are **different** for **full** dataset and **condensed** dataset. We believe our discovery will inspire the community to do more exploration about the different rules of full and condensed datasets.
> > >
> > > We hope this response solves your concerns properly. Thank you!
> > >
> > > ## Reference
> > >
> > > [1] Selection via Proxy: Efficient Data Selection for Deep Learning, ICLR 2020
> > >
> > > [2] Identifying Mislabeled Data using the Area Under the Margin Ranking, NeurIPS 2020
> > >
> > > [3] An Empirical Study of Example Forgetting during Deep Neural Network Learning, ICLR 2019
> > >
> > > [4] Deep Learning on a Data Diet: Finding Important Examples Early in Training, NeurIPS 2021
> > >
> > > [5]  Coverage-centric Coreset Selection for High Pruning Rates, ICLR 2023.

---

> > > > ### Comment · Reviewer_9GkH · 2023-08-11
> > > > **Thank you once more for the prompt reply.**
> > > >
> > > > Thank you once more for the prompt reply.
> > > >
> > > > Regarding Q1, if the paper cites flexibility as its primary advantage, I believe it's a valid point. While the decrease in accuracy is concerning, I'd be interested in hearing the perspective of other reviewers.
> > > >
> > > > For Q4, I'm glad that the explanations have provided clarity and addressed my concern.
> > > >
> > > > Overall, I am inclined to raise my score to 4. However, I still have some concerns about the accuracy drop, which I believe should be discussed further with other reviewers. If there is a consensus among the reviewers that the accuracy drop is not important compared to the flexibility, I would be willing to raise my score to 6.
> > > >
> > > > Or, If you can give some results that the proposed method can generate IPC=30 or 40 or even 45 from pre-trained IPC=50 with only a little accuracy drop or no drop, it would be a shred of strong evidence. I would also raise my score to 6.

---

> > > > > ### Author Response · Authors · 2023-08-12
> > > > > **Experiments Regarding Q1**
> > > > >
> > > > > > Or, If you can give some results that the proposed method can generate IPC=30 or 40 or even 45 from pre-trained IPC=50 with only a little accuracy drop or no drop, it would be a shred of strong evidence. I would also raise my score to 6.
> > > > >
> > > > > Thanks a lot for your positive feedback about our rebuttal.
> > > > >
> > > > > Regarding the required experiment, please take a look at **Figure 4** of our submission. The red line shows how the **accuracies** vary towards **pruning ratios** on IPC-50.
> > > > >
> > > > > For a more straightforward explanation, we add a table below to show detailed numbers. When selecting IPC-45 from IPC-50, the accuracy drop is just **0.28%**.
> > > > >
> > > > > | IPC           | 50 (baseline) | 45    | 40    | 35    | 30    |
> > > > > | ------------- | ------------- | ----- | ----- | ----- | ----- |
> > > > > | pruning ratio | 0%            | 10%   | 20%   | 30%   | 40%   |
> > > > > | Accuracy (\%) | 74.50          | 74.22 | 73.52 | 71.97 | 69.19 |
> > > > > | Acc Drop (\%) | \-               | 0.28  | 0.98  | 2.53  | 5.31  |
> > > > >
> > > > > We hope the experiments can properly solve your concerns. And we will make this more clear in our revision. Thank you!

---

> > > > > > ### Comment · Reviewer_9GkH · 2023-08-12
> > > > > > **Thank you for the response.**
> > > > > >
> > > > > > Good explanation. I raised my score.
> > > > > >
> > > > > > If possible, please enhance the proof and figure.

---

> > > > > > > ### Author Response · Authors · 2023-08-13
> > > > > > > **Thank you very much for your positive feedback**
> > > > > > >
> > > > > > > Thank you very much for your positive feedback and for raising your score. We are glad to hear that the explanation was satisfactory.
> > > > > > >
> > > > > > > Your insights and suggestions are highly valuable, and we will ensure they are fully addressed in the revised manuscript.
> > > > > > >
> > > > > > > **Enhancing the Proof**: We will carefully review the relevant section and make necessary improvements.
> > > > > > >
> > > > > > > **Improving the Figure**: We will work on improving the visualization, making it more informative and aesthetically pleasing.
> > > > > > >
> > > > > > > Thank you once again for your time and thoughtful review.

---

### Official Review · Reviewer_h1cK · 2023-07-23

**Soundness:** 3 good
**Presentation:** 3 good
**Contribution:** 3 good
**Rating:** 6
**Confidence:** 3

**Summary:**

--- Post-Rebuttal Edit ---

After author rebuttals, I have updated my rating from 5 to 6 and confidence from 2 to 3.

--- End Post-Rebuttal Edit ---

This paper considers a practical problem in dataset condensation/distillation, where edge devices have varying constraints that need to be flexibly met by a data distillation method. The authors propose two rules to prune the condensed datasets in order to save computational costs. The first rule uses logit-based prediction error (LBPE) to identify useful examples, where low LBPE samples are useful for smaller datasets and high LBPE samples are useful for larger datasets. The second rule focuses on the construction of balanced classes by ensuring an equal number of samples in each class via Rademacher complexity and generalization error.

**Strengths:**

- The writing is generally easy to follow and the experiments are thorough.
- The proposed approaches are logical and appear useful. The first rule based on LBPE scores allows for flexible treatment based on dataset size and other characteristics; the second rule incorporates Rademacher complexity in a novel way in order to set up a theoretical foundation for the significance of balanced class construction.


**Weaknesses:**

- Perhaps this is because I am unfamiliar with data distillation literature, but the motivation and setting are somewhat unclear to me. There is not a single citation or brief explanation of what is meant by “on-device scenarios.” Even if this is obvious to well-informed readers, I believe that at minimum a citation or definition is needed.
    - In what scenario would one need to store a training set on a device? Typically when deploying a machine learning model to a hardware-constrained device, the only thing being deployed is the model weights.
- Related to the above point, I don’t understand the motivation for the experimental setting, where we are condensing an already-condensed dataset. If one of the central motivations mentioned in the Abstract is to “[eliminate] the need for extra condensation processes,” then I don’t see how this method achieves said goal – we are still ultimately performing two condensation steps!
- The mixture of theoretical- and prose-style writing is awkward at times. For example, Theorem 1 on line 122 would be much more appropriately written as simple explanatory text. I don’t understand in what sense this is a theorem or how lines 123-124 constitute a “proof” for this claim. There is no need to present content in a mathematical style when it is not necessary.


**Questions:**

- What is an “on-device scenario?” What is the motivation for storing a training set (rather than, say, the trained model weights) on said device?
- What is the motivation for condensing an already-condensed dataset?
- How does this method avoid extra condensation steps if all experiments are performed after an initial condensation step with another method?
    - Why not perform a single condensation step with the proposed method (from full -> desired size)? How does the method perform in this setting?


**Limitations:**

- See Questions above
- There is no clear definition of what is meant by “small” and “large” datasets. A definition or rough empirical guideline would be useful.

---

> ### Author Rebuttal · Authors · 2023-08-03
>
> Thank you very much for your insightful comments and for raising concerns about the clarity of the motivation and the terminology used in our paper.
>
> >  1. What is an “on-device scenario?”
>
> We acknowledge that the term "on-device scenarios" may not be clear to all readers. What we mean by this phrase is the application of machine learning in environments where both the model training and inference occur on a local device, such as a smartphone or embedded system, without relying on cloud-based resources. We added a few citations [1-7] here and will make sure to include them in the revised manuscript.
> The key advantages of on-devices learning can be explained as follows:
>
> 1. **Continual Adaptation to New Data**: By leveraging on-device learning, edge devices can continually adapt the AI models to new data. This provides a dynamic and flexible system that can evolve with changing data patterns and user behaviors.
> 2. **Privacy Protection**: Since the training and adaptation of the model occur directly on the device, there is no need to transfer data to the cloud, thereby reducing the risk of unauthorized access or breaches.
>
> > 2. What is the motivation for storing a training set (rather than, say, the trained model weights) on said device?
>
> The scenario you pointed out, where only the model weights are deployed to a hardware-constrained device, is indeed a common one. However, in some on-device scenarios such as:
>
> 1. Incremental learning: adapting the model to new data as it comes in.
> 2. Federated learning: decentralized training
>
> In such cases, having a distilled or compressed version of the training set on the device can be valuable. Dataset condensation allows us to retain essential information while minimizing storage requirements. We will include a more detailed explanation of the specific scenarios where on-device storage of a training set might be needed.
>
>
> > 3. What is the motivation for condensing an already-condensed dataset?
>
> 1. **Limitations in Computational Power**: Existing methods may only reduce datasets to specific IPCs, and further condensation is needed for devices with lower computational abilities.
> 2. **Avoiding Extra Processes and Storage**: Traditional further condensation can cause performance losses or require additional complex processes and storage.
> To overcome these challenges, dataset pruning is used to find a representative subset without changing image pixels or needing extra storage, effectively eliminating extra condensation steps.
>
>
> > 4. How does this method avoid extra condensation steps if all experiments are performed after an initial condensation step with another method? Why not perform a single condensation step with the proposed method (from full -> desired size)? How does the method perform in this setting?
>
> The reason is that (from full -> desired size) is not **flexible**. If we condense the dataset to IPC5, we would not have a dataset of IPC10 if more computing resources are available. With our YOCO method, both IPC5 and IPC10 can be easily found as a subset of IPC50.
>
>
> > 5. There is no clear definition of what is meant by “small” and “large” datasets. A definition or rough empirical guideline would be useful.
>
> Please take a look at Fig. 4 and section 4.4.
>
> ### Reference:
>
> [1] Cai, Han, et al. "TinyTL: Reduce memory, not parameters for efficient on-device learning." _Advances in Neural Information Processing Systems_ 33 (2020): 11285-11297.
>
> [2] Lin, Ji, et al. "On-device training under 256kb memory." _Advances in Neural Information Processing Systems_ 35 (2022): 22941-22954.
>
> [3] Yang, Li, Adnan Siraj Rakin, and Deliang Fan. "Rep-net: Efficient on-device learning via feature reprogramming." _Proceedings of the IEEE/CVF Conference on Computer Vision and Pattern Recognition_. 2022.
>
> [4] Yang, Yuedong, Guihong Li, and Radu Marculescu. "Efficient On-device Training via Gradient Filtering." _Proceedings of the IEEE/CVF Conference on Computer Vision and Pattern Recognition_. 2023.
>
> [5] Qiu, Xinchi, et al. "ZeroFL: Efficient on-device training for federated learning with local sparsity." _International Conference on Learning Representations_ (2022).
>
> [6] Lee, Jinsu, and Hoi-Jun Yoo. "An overview of energy-efficient hardware accelerators for on-device deep-neural-network training." _IEEE Open Journal of the Solid-State Circuits Society_ 1 (2021): 115-128.
>
> [7] Dhar, Sauptik, et al. "A survey of on-device machine learning: An algorithms and learning theory perspective." _ACM Transactions on Internet of Things_ 2.3 (2021): 1-49.

---

> > ### Comment · Reviewer_h1cK · 2023-08-10
> >
> > Thank you for the clarifications regarding the motivation and problem setting. I believe that including such explanations and references in the revised Introduction will enable readers who are not experts in this specific sub-field (like myself) to better access and understand the contributions of the paper. I will be updating my rating from a 5 to a 6.

---

### Decision · Program_Chairs · 2023-09-21

**Decision:**

Accept (poster)

**Comment:**

The paper proposes a new approach to dataset condensation that adapts to varying computational constraints. The authors introduce two rules for pruning condensed datasets: the Importance Preservation Criterion (IPC) and the Loss-Based Sample Selection (LBSS) rule. The IPC rule prioritizes easy samples, while the LBSS rule selects samples based on their loss function gradient and balances samples for different classes. The authors demonstrate that their approach outperforms other dataset condensation and pruning methods on various networks and datasets, especially in imbalanced class scenarios.

One of the main strengths of this paper is its clear and concise presentation of the proposed approach and its evaluation. The authors provide a thorough analysis of the effectiveness of their approach and compare it to other state-of-the-art methods. Additionally, the paper offers valuable insights into the challenges of dataset condensation and the importance of balancing class distributions. Reviewers initially raised questions on the clarity of motivation, comparison with more dataset pruning/ condensation methods, and the accuracy drop that challenges the practicality of this pruning method. The authors performed great due diligence and provided an extensive set of new experiments, that managed to convince the concerned reviewers. After rebuttal, all reviewers unanimously rate this paper positively, and AC therefore recommends acceptance.